# The pharmacoepigenomic landscape of cancer cell lines reveals the epigenetic component of drug sensitivity

Alexander Joschua Ohnmacht[1,2], Anantharamanan Rajamani[3,4,5], Göksu Avar [1,2], Ginte Kutkaite [1,2], Emanuel Gonçalves[6,7], Dieter Saur [3,4,5] & Michael Patrick Menden [1,2,8 ✉]

Aberrant DNA methylation accompanies genetic alterations during oncogenesis and tumour homeostasis and contributes to the transcriptional deregulation of key signalling pathways in cancer. Despite increasing efforts in DNA methylation profiling of cancer patients, there is still a lack of epigenetic biomarkers to predict treatment efficacy. To address this, we analyse 721 cancer cell lines across 22 cancer types treated with 453 anti-cancer compounds. We systematically detect the predictive component of DNA methylation in the context of transcriptional and mutational patterns, i.e., in total 19 DNA methylation biomarkers across 17 drugs and five cancer types. DNA methylation constitutes drug sensitivity biomarkers by mediating the expression of proximal genes, thereby enhancing biological signals across multi-omics data modalities. Our method reproduces anticipated associations, and in addition, we find that the *NEK9* promoter hypermethylation may confer sensitivity to the NEDD8-activating enzyme (NAE) inhibitor pevonedistat in melanoma through downregulation of *NEK9*. In summary, we envision that epigenomics will refine existing patient stratification, thus empowering the next generation of precision oncology.

[1] Computational Health Center, Helmholtz Munich, 85764 Neuherberg, Germany. [2] Department of Biology, Ludwig-Maximilians University Munich, 82152 Martinsried, Germany. [3] Division of Translational Cancer Research, German Cancer Research Center (DKFZ) and German Cancer Consortium (DKTK), Im Neuenheimer Feld 280, 69120 Heidelberg, Germany. [4] Chair of Translational Cancer Research and Institute of Experimental Cancer Therapy, Klinikum rechts der Isar, School of Medicine, Technische Universität München, Ismaninger Str. 22, 81675 Munich, Germany. [5] Center for Translational Cancer Research (TranslaTUM), School of Medicine, Technical University of Munich, Ismaninger Str. 22, 81675 Munich, Germany. [6] Instituto Superior Técnico (IST), Universidade de Lisboa, 1049-001 Lisbon, Portugal. [7] INESC-ID, 1000-029 Lisbon, Portugal. [8] Department of Biochemistry and Pharmacology, University of Melbourne, Victoria, VIC 3010, Australia. ✉email: michael.menden@helmholtz-munich.de

Precision oncology adverts to stratifying patients based on tumour entities and their molecular profiles to enhance drug efficacy and reduce toxicity[1]. The success rate of clinical trials without a molecular biomarker is estimated to be 1.6% and is increased to 10.7% when using an appropriate patient stratification[2]. Accordingly, methods that identify biomarkers and thereby facilitate clinical translation are crucial for the rapid development of novel cancer treatments.

In human tumours, aberrant DNA methylation has been shown to deregulate oncogenic pathways[3] and to contribute to the acquisition of drug resistance[4,5]. For example, DNA methylation in promoter, enhancer and CpG island regions has revealed epigenetic mechanisms involved in the transcriptional activity of several key cancer genes[3,6]. In particular, the downregulation of tumour suppressor genes by hypermethylation of CpG sites in gene promoters is a hallmark of many cancer types[7]. In contrast, the hypermethylation of CpG sites in gene bodies is often reported to be positively correlated with gene expression[8].

Molecularly characterised cancer cell lines are a useful and scalable model system for drug discovery[9]. They have empowered large high-throughput drug screens (HTS)[10–15], which include cell line panels of >1000 cell lines and are aimed to characterise the biomarker landscape of cancer[16]. For example, skin cutaneous melanoma cell lines (SKCM) harbouring BRAF V600E mutations are vulnerable to BRAF kinase inhibitors, and furthermore, this in vitro observation generalises to in vivo models and melanoma patients[17]. Genetic alterations are the causally related disease aetiology of cancer. Thus, most molecular biomarker studies have focused on somatic mutations and copy number variations. However, despite the growing utility of epigenetic biomarkers in clinics and an increasing number of commercially available diagnostic tests involving DNA methylation[18], prognostic and predictive epigenetic biomarkers are still sparse[19].

Few efforts have been dedicated to identifying DNA methylation biomarkers of drug response. For example, DNA methylation has been used to identify the CpG island methylator phenotype (CIMP)[20]. It has previously been suggested as a predictive biomarker[21], however, its definition is still inconsistent[22], challenging to mechanistically interpret and limited to a handful of cancer types[20,23,24]. Furthermore, predictive DNA methylation biomarkers in HTS are commonly assessed by summarising CpG sites in promoters and CpG islands[11,21]. For these summarised regions, machine learning models have been used to predict drug response[25,26] of preselected genes involved in DNA methylation or demethylation[26]. In summary, these methods either do not leverage the full epigenome on the CpG site resolution, build evidence in multi-omics data modalities across different datasets, or lack mechanistic interpretations.

In order to empower epigenetic response biomarkers, our objectives were: (1) Identify DNA methylation regions associated with drug response in HTS; (2) Integrate genetic, epigenetic and transcriptomic data modalities of cancer cell lines for increasing evidence and interpretability; (3) Verify these epigenetic regulations of gene expression in human primary tumours and thus enhancing clinical translatability; (4) Finally, map the epigenetically regulated genes onto protein-protein signalling networks, and link them to their respective drug targets, thereby obtaining interpretable, actionable and translatable mechanisms. Our systematic analysis of the pharmacoepigenomic landscape in HTS, accompanied by thorough filtering for layer-wise evidence, interpretability and translatability, may pave the way for epigenetic response biomarkers in cancer.

## Results

For the discovery of DNA methylation biomarkers of drug response, we analysed methylation patterns of 721 cancer cell lines from 22 cancer types treated with 453 anti-cancer compounds. The data was derived from the Genomics of Drug Sensitivity in Cancer (GDSC; Fig. 1a) project[11], which has since expanded its set of screened compounds compared to the original publication[27,28]. Drug responses of cancer cell lines were characterised by their area under the drug response curve (AUC; Fig. 1b), for which low AUC values convey high sensitivity to the respective compound.

We first systematically searched for methylation regions with differential drug response in cancer cell lines, i.e., drug differentially methylated regions (dDMRs) by adaptively grouping spatially correlated CpG sites contained in the Infinium HumanMethylation450 BeadChip array (Fig. 1c; Methods). Secondly, we filtered for dDMRs which may mediate proximal gene expression (Fig. 1d; Methods), which thereby increases evidence of functional epigenetic events impacting drug response (Fig. 1e). Subsequently, we filtered for concordantly observed epigenetic mechanisms in human primary tumour samples from The Cancer Genome Atlas (TCGA; Fig. 1f; Methods), which yielded a prioritisation list of tumour-generalisable dDMRs, (tgdDMRs). Lastly, we correlated tgdDMRs with somatic mutations in cancer genes (Fig. 1g) and used shortest path algorithms applied to protein-protein interaction networks (Fig. 1g, h; Methods) to derive relationships between drug targets and proximal tgdDMR genes encoding respective proteins to support tgdDMRs further. In total, we found 19 tgdDMRs, i.e., predictive epigenetic biomarkers of drug response.

### Identification of epigenetic drug response biomarkers from high-throughput drug screens.

Analysing the DNA methylation and gene expression profiles of cancer cell lines stemming from 22 cancer types highlighted that the variance within cancer types is lower compared to the variance between cancer types (Fig. 2a and Supplementary Fig. 1a). Hence, we stratified cell lines into cancer types for subsequent modelling. For each cancer type and screened compound, we employed linear models and called drug differentially methylated regions (dDMRs; Methods), i.e., regions for which the methylation in CpG sites associates with drug response quantified by AUC. In total, we identified 802 dDMRs for 186 drugs in 22 cancer types (dDMR calling, adj. $p < 10^{-6}$; Fig. 2b and Supplementary Fig. 1b). We observed a linear relationship between the amount of found dDMRs and the sample size of the investigated cancer type (Pearson's $r = 0.81$, $p = 5.1 \times 10^{-6}$, correlation test; Supplementary Fig. 1c).

The distribution of significant drugs across cancer types was heterogeneous, but we identified enrichments of drug classes between cancer types (one-sided hypergeometric test, FDR < 0.05; Supplementary Data 1): Drugs that target the ERK-MAPK signalling pathway (trametinib, PD0325901, ulixertinib, selumetinib, VX-11e and CI-1040) were enriched in colorectal cancer (COREAD, odds ratio = 6.3), drugs that target EGFR signalling (afatinib, sapitinib, AZD3759, erlotinib, gefitinib and pelitinib) were enriched in lung adenocarcinoma (LUAD, odds ratio = 15.0) and drugs that are involved in targeting mitosis (alisertib, vinblastine, vinorelbine, GSK1070916, epothilone B, docetaxel, ARRY-520, S-trityl-L-cysteine) were enriched in small-cell lung cancer (SCLC, odds ratio = 4.9).

The distribution of CpG site counts per dDMR had a median of seven sites per dDMR. Furthermore, 132/802 dDMRs comprised >10 CpG sites, whilst 147 dDMRs contained <5 sites (Supplementary Fig. 1d). dDMRs were enriched for DNAase I hypersensitive sites (DHS, $p < 10^{-16}$, odds ratio = 3.32, one-sided

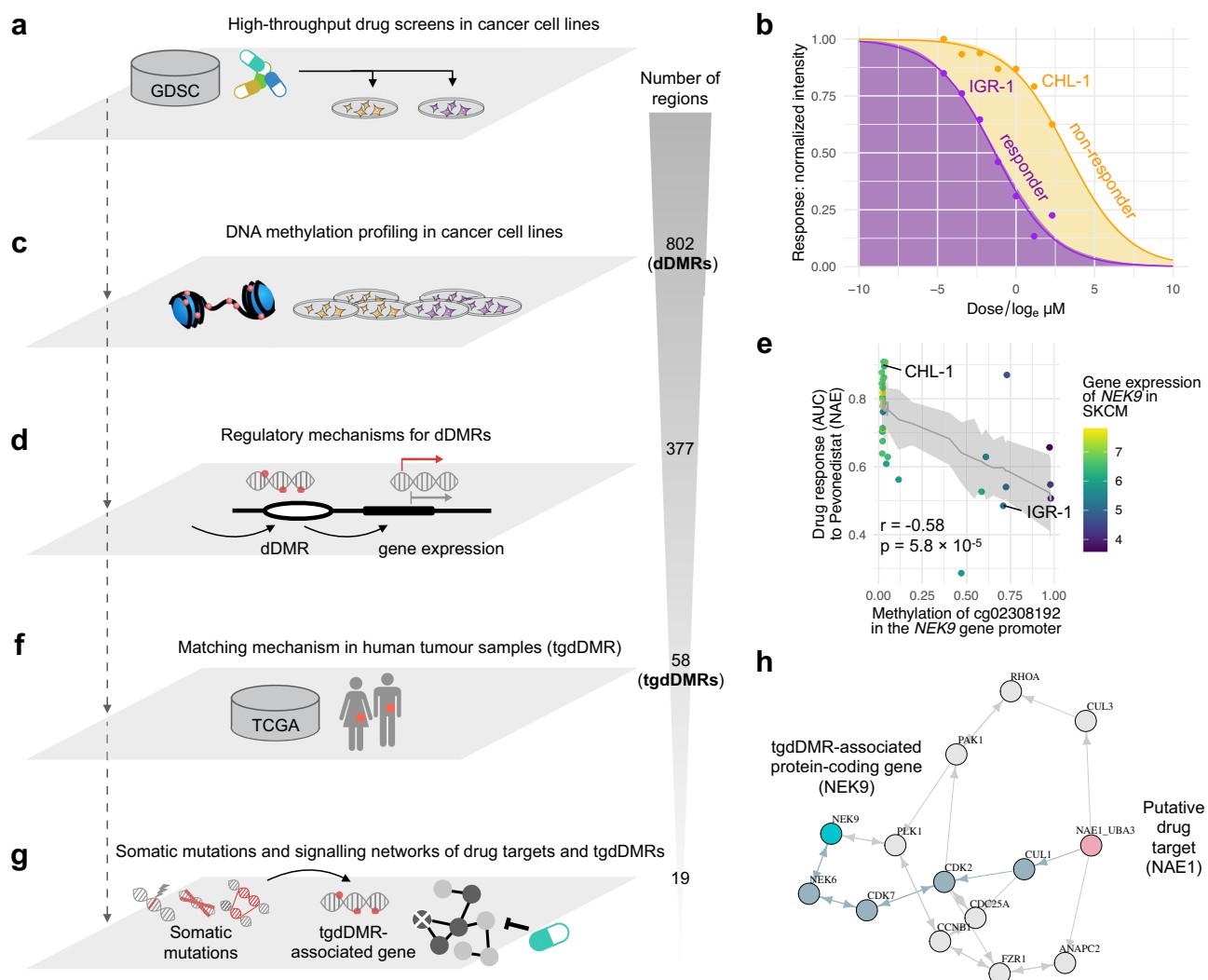

**Fig. 1 Analysis workflow for the identification of epigenetic biomarkers of drug response. a** The Genomics of Drug Sensitivity in Cancer (GDSC) project contains 721 cancer cell lines from 22 cancer types, which were epigenetically characterised and screened across 453 compounds. **b** The dose-response curves of a responder and non-responder melanoma cell line treated with pevonedistat. **c** We identified 802 drug differentially methylated regions (dDMRs). **d** The set of dDMRs is filtered for regulatory mechanisms, i.e., correlated gene expression of proximal genes, resulting in 377 functionally interpretable dDMRs. **e** For example, the dDMR in the *NEK9* promoter is associated with the expression of *NEK9* and is additionally correlated with drug response to pevonedistat. The error bars corresponding to 95% confidence intervals, the raw *p*-value (*p*) for the respective CpG site and the Pearson correlation coefficient (*r*) are displayed. **f** In total, the methylation of 58 epigenetic biomarkers of drug response were observed to be consistently correlated with the expression of their proximal gene in TCGA primary tumours. **g** The set of tgdDMRs was investigated for correlated somatic mutations in cancer cell lines. Additionally, for gaining further mechanistic insights, shortest-path algorithms traversed protein-protein signalling networks containing tgdDMR-associated genes as well as the respective drug targets and revealed additional evidence for 19 tgdDMRs. **h** The predictive biomarker *NEK9* (light blue) is connected within five steps to the drug target of pevonedistat, i.e., the NEDD8-activating enzyme NAE (pink). In the graph, nodes that are traversed with a shortest path are highlighted by the blue-grey colour among the alternative paths. The used human icons are from the AIGA symbol signs collection and are in the public domain.

Fisher's test; Fig. 2c, d) and sites in CpG islands ($p < 10^{-16}$, odds ratio = 3.13, one-sided Fisher's test; Fig. 2c, d). Furthermore, we investigated dDMRs in proximity of cancer genes based on annotations of the Network of Cancer Genes (NCG) project[29]. DNA Methylation sites on the 450k microarrays have higher seeding density in the vicinity of cancer genes, i.e., 645/674 (96%) of cancer genes contained >10 profiled CpG sites compared to 16,213/20,557 (79%) of non-cancer genes. To alleviate this bias, we only tested genes with at least ten proximal CpG sites, which resulted in 16,858 background genes and 645/16,858 (3.8%) cancer genes. We observed 503 genes in proximity to identified dDMRs, of which 27 were cancer genes (5.4%; Supplementary Fig. 1e), thus cancer genes were significantly enriched ($p = 0.049$,

odds ratio = 1.44, one-sided Fisher's test). The most prevalent cancer genes were *APC* and *SKI* found across two cancer types. For reference, the most prevalent non-cancer genes were *PTPRN2* and *DKK1*, which were found in five and four cancer types, respectively (Supplementary Data 2).

Among the cancer genes associated with dDMRs, we found that *MGMT* dDMR methylation in low-grade glioma was associated with response to JQ1 (BET inhibitor, dDMR calling, adj. $p < 10^{-6}$; Supplementary Fig. 1f). The epigenetic silencing of *MGMT* is frequently debated as a clinical biomarker[30] and previous work revealed that JQ1 disturbs DNA damage responses by attenuating *MGMT* expression in glioblastoma cells[31]. While the different treatment responses are often attributed to somatic

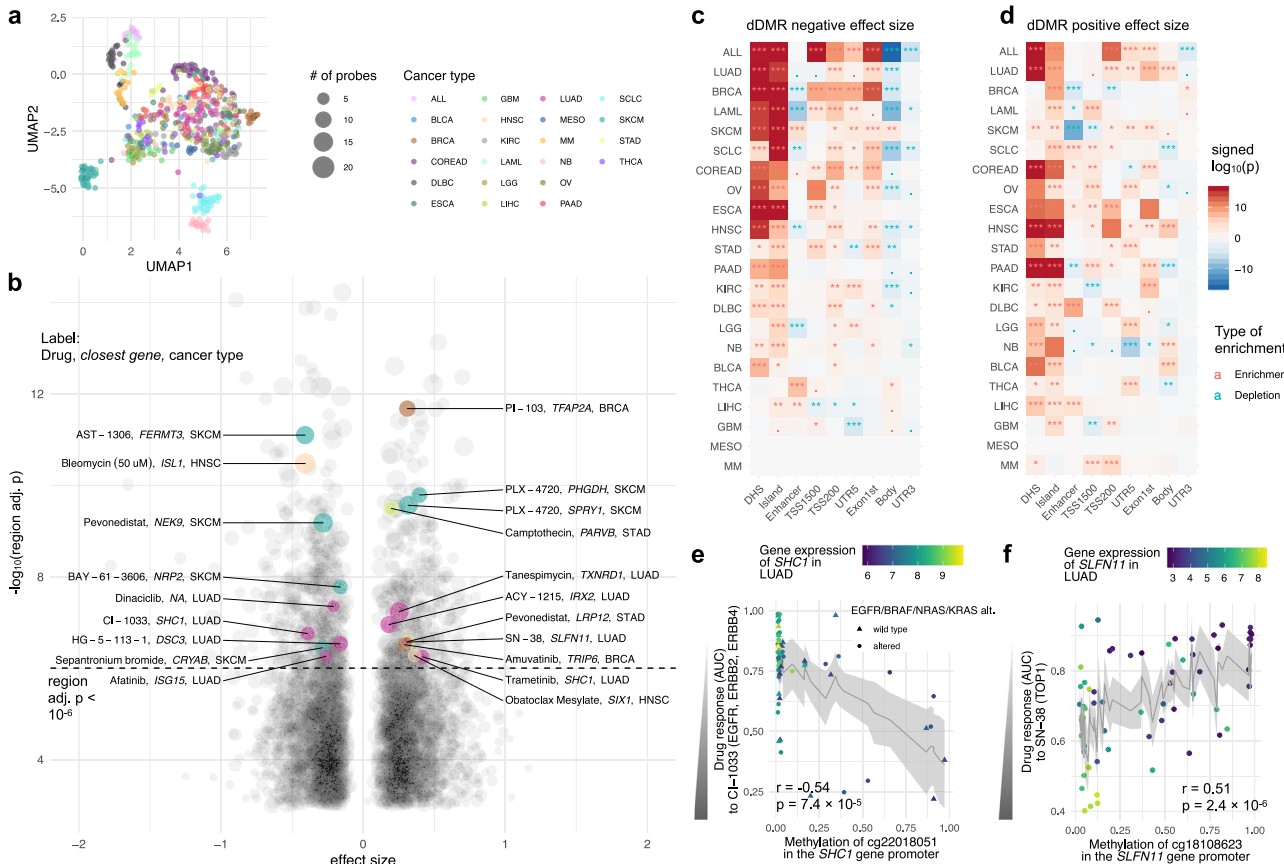

**Fig. 2 Heterogeneity of epigenetic patterns across cancer types results in a rich resource of dDMRs. a** Cancer type specific pattern of DNA methylation profiles of cancer cell lines in the GDSC. **b** Significant dDMRs across 22 cancer types and 186 drugs. The size of the data points indicates the amount of CpG sites in the identified dDMR. Genomic regions are labelled by the gene name in the closest vicinity. The enrichment of functional genomic regions in dDMRs is visualised in heatmaps for the scenario in which **c** hypermethylation confers drug sensitivity or **d** hypomethylation confers sensitivity. We tested enrichments for: genomic regions in the form of DNAaseI hypersensitive sites (DHS), CpG sites within CpG islands, enhancer regions, regions within 200 and 1500 bases upstream of the transcription start site (TSS200 and TSS1500), the 5' untranslated region (UTR5), the 1st exon, gene body and 3' untranslated region (UTR3). **e** The association between *SHC1* promoter hypermethylation and CI-1033 response in LUAD; and **f** the association between *SLFN11* gene hypomethylation and response to SN-38. The error bars corresponding to 95% confidence intervals, the raw *p*-value (*p*) for the respective CpG site and the Pearson correlation coefficient (*r*) are displayed.

mutations in cancer genes, this suggests that DNA methylation can function as a complementary mechanism.

A negative effect size of a dDMR indicates that drug-sensitive cell lines are hypermethylated. Here, this is exemplified by the methylation status of *SHC1*, which was found to be associated with the EGFR, ERBB2 and ERBB4 inhibitor CI-1003 in LUAD (Fig. 2e). We observed that LUAD cell lines with a hypermethylated promoter of *SHC1* showed lower expression levels and were more sensitive to CI-1003 (Fig. 2e). Indeed, the adaptor protein SHC1 is involved in promoting the downstream signalling of ERK through EGFR[32]. No correlations between *SHC1* methylation and alterations in the ERK signalling pathway such as *EGFR*, *BRAF*, *NRAS* or *KRAS* mutations or amplifications were found. Clinical trials have reported benefits for non-small cell lung cancer patients with EGFR mutant tumours treated with the pan-ERBB inhibitor dacomitinib[33,34]. Thus, *SHC1* silencing through DNA hypermethylation may be a sufficient but not necessary condition for sensitivity to ERBB inhibitors.

Overall, CpG sites in gene promoters were particularly enriched in dDMRs with a negative effect size ($p < 10^{-15}$, one-sided Fisher's test; Fig. 2c). For dDMRs with a negative effect size, methylation sites were usually hypomethylated across cancer cell lines, with a few treatment-sensitive cell lines that were hypermethylated (Supplementary Fig. 2).

In contrast to above, dDMRs with positive effect size contained methylated CpG sites that were mostly distributed across diverse genomic locations (Fig. 2d) and their hypomethylation was associated with drug sensitivity (Supplementary Fig. 2). Furthermore, we found enrichments of dDMRs with positive effect size within 200 bases upstream of the transcription start site (TSS200) for 11/22 cancer types ($p < 0.001$, one-sided Fisher's test; Fig. 2d). Exemplifying a dDMR with positive effect size, the hypomethylation of the *SLFN11* promoter was significantly associated with sensitivity to SN-38 in LUAD (Fig. 2f). The topoisomerase I inhibitor SN-38, the active metabolite of irinotecan, inhibits DNA replication through binding to the topoisomerase I-DNA complex and thus promotes DNA double-strand breaks. *SLFN11* is a putative DNA/RNA helicase that sensitises cancer cells to DNA damaging agents by killing cells with defective DNA repair[35]. Its expression has been discussed extensively as a predictive biomarker for compounds targeting the DNA damage response[36,37]. Here, we show that cells with hypomethylated *SLFN11* show high *SLFN11* expression and sensitivity to SN-38.

For validating dDMRs, we retrieved independent drug response data from the Cancer Therapeutics Response Portal (CTRP; Methods). We found that 236/802 dDMRs (29.4%) had overlapping data on cancer cell lines and drugs between GDSC

and CTRP. Among these, 193/236 (81.8%) had consistent effect size (Supplementary Fig. 3a), with an overall correlation of Pearson's $r = 0.46$ ($p = 9.7 \times 10^{-14}$, correlation test; Supplementary Fig. 3b). Furthermore, we validated our dDMRs with independent methylation data, i.e., reduced representation bisulfite sequencing for DNA methylation profiling (RRBS; Methods) extracted from the Cancer Cell Line Encyclopaedia (CCLE). This only reduced the overlapping data of dDMRs slightly to 227/802 (28.3%), and 164/227 (72.2%) of these dDMRs displayed consistent effect size (Supplementary Fig. 3a), with a correlation of Pearson's $r = 0.43$ ($p = 1.2 \times 10^{-11}$, correlation test; Supplementary Fig. 3c), highlighting the ability of our method to yield reproducible results for independent drug screenings and DNA methylation experiments.

**Epigenetic biomarkers interpreted through gene regulatory mechanisms.** Hypermethylation of promoter regions is an established mechanism to reduce sufficient transcription factor binding and regulate gene expression accordingly[38]. Thus, most methylation biomarker discovery efforts focus on gene promoter regions and neglect other regulatory mechanisms[11,21,25,26]. For example, the deregulation of methylation patterns in gene bodies was also reported to alter gene expression profiles[8]. In order to address this, we generalised our working hypothesis and explored the DNA methylation of any dDMR that may mediate gene expression of proximal genes (Methods).

Upon systematic analysis with the Enhancer Linking by Methylation/Expression Relationships (ELMER) method[39], we observed that 377/802 dDMRs (47.0%) showed at least one significantly associated gene in the proximity of its genomic region (emp. adj. $p < 0.001$; Methods). In total, 576 genes were associated with these 377 dDMRs. For each gene associated with a dDMR, we independently correlated its expression and drug response with a linear model fit (Fig. 3a–d). In summary, we observed four distinct mechanisms which may drive drug sensitivity, i.e., hypermethylation with either downregulated gene expression (Case 1, $n = 216$; Fig. 3a) or upregulated gene expression (Case 2, $n = 110$; Fig. 3b), and hypomethylation with either upregulated gene expression (Case 3, $n = 162$; Fig. 3c) or downregulated gene expression (Case 4, $n = 88$; Fig. 3d). We exemplified each case in cancer cell lines and their mechanistic consistency in primary tumours (Fig. 3e–l).

For both Cases 1 and 2, hypermethylated dDMRs were associated with drug sensitivity (negative effect size in Fig. 2b). The majority of dDMRs belonged to Case 1, which was distinguished by promoter regions (Fig. 3a). It resembles the canonical mechanism in which hypermethylation of promoter regions downregulates the expression of their associated proximal gene and thereby confers drug sensitivity. This behaviour is exemplified by the methylation of the *SHC1* promoter and its gene expression in LUAD cell lines (Fig. 3e). Additionally, we verified the association of the epigenetic status and gene expression in LUAD human tumour samples (Fig. 3f).

For Case 2, hypermethylation of dDMRs correlated with higher expression of proximal genes (Fig. 3g, h). This is a less frequent epigenetic regulation mechanism, however, it is consistent with previous studies reporting both behaviours[8,40–42]. As an example, the hypermethylation of the *OPLAH* dDMR was associated with the upregulation of *OPLAH* expression in SKCM cancer cell lines and HG-6-64-1 drug sensitivity (Fig. 3g). In addition, this epigenetic regulation of *OPLAH* expression was also demonstrated in primary tumour samples (Fig. 3h).

Cases 3 and 4 were characterised by hypomethylated dDMRs that were associated with drug sensitivity (positive effect size in Fig. 2b), which could also be distinct by negative or positive

correlations of dDMRs with gene expression for Case 3 and Case 4, respectively. For example, we found that the hypomethylation of the *SLFN11* dDMR in LUAD was associated with higher *SLFN11* expression (Fig. 3i), which was further verified in human tumour samples (Fig. 3j). In contrast, the hypomethylation of *PITX2* dDMR was linked to teniposide drug sensitivity, however, the hypermethylation of *PITX2 dDMR* was positively associated with *PITX2* expression in cancer cell lines and human tumour samples (Fig. 3k, l).

In summary, drug sensitivity in cancer cell lines may be driven by either hypermethylation (Cases 1 and 2) or hypomethylation (Cases 3 and 4) of dDMRs and can either present negatively correlated gene expression (Cases 1 and 3) or positively correlated gene expression (Cases 2 and 4). Case 1 has been the focus of most epigenetic biomarker studies, whilst we systematically investigated all 4 cases (Supplementary Data 2) and therefore can provide broader mechanistic insights.

**Epigenetic and transcriptional mechanisms in primary tumours increase evidence of drug response biomarkers.** In the section above, we highlighted four distinct epigenetic mechanisms that may drive drug response, i.e., Case 1-4. Each of them was exemplified in cancer cell lines (Fig. 3e, g, i, k), and consecutively, further supported by concordant methylation and proximal gene expression patterns in tumours (Fig. 3f, h, j, l). Here, we systematically assessed all 377 short-listed dDMRs from above, to investigate concordant epigenetic regulation patterns in primary tumours leveraging ELMER[39] also in TCGA tumour samples[43] (Methods). In total, we investigated a subset of 241/377 dDMRs for which the associated cancer type data was available in TCGA. We observed that 58/241 (24.1%) of dDMRs showed a significant association with their proximal genes in tumours (ELMER, emp. adj. $p < 0.001$; Methods). We called this selection of epigenetic biomarkers tumour-generalisable dDMRs (tgdDMRs). For the final selection, we found 19/58 tgdDMRs for which the protein encoded by the associated gene was connected to the corresponding drug targets in the protein-protein signalling network OmniPath[44] (Methods). These 19 tgdDMRs (Supplementary Data 2) contained proposed biomarkers for 17 anti-cancer drugs across five cancer types (Fig. 4a), i.e., LUAD $n = 7$ (Supplementary Fig. 4), SKCM $n = 6$ (Supplementary Fig. 5), breast cancer (BRCA) $n = 2$ (Supplementary Fig. 6), head and neck cancer (HNSC) $n = 2$ (Supplementary Fig. 6), and stomach adenocarcinoma (STAD) $n = 2$ (Supplementary Fig. 6).

We found that the majority of tgdDMRs (15/19) were in promoter regions, which is concordant with previous computational strategies that focused solely on promoters to identify epigenetic response biomarkers. However, the remaining 4 tgdDMRs, which constitute >20% of our identified lead biomarkers, had distinctly different epigenetic regulation mechanisms, i.e., were located in either the gene body or distal regions (Fig. 4b). In addition, we found that all tgdDMRs had negative correlations with a proximal gene, which correspond to mechanism Case 1 or Case 3 (Fig. 3a, c). Furthermore, for 10/19 tgdDMRs the expression of proximal genes in cell lines itself was independently associated with drug response in cancer cell lines ($p < 0.05$, linear model fit; Methods), thus having a functional interpretation across two molecular layers.

For additional evidence of tgdDMRs, we again leveraged the CTRP and CCLE datasets as validation cohorts. For the tgdDMRs that had overlapping drug response data, we found that 7/9 tgdDMRs showed consistent effect sizes in the CTRP screen, with an increased correlation of Pearson's $r = 0.75$ ($p = 0.02$, correlation test; Fig. 4c) compared to unfiltered dDMRs in the previous section. Additionally, 5/7 of the tgdDMRs overlapping with the

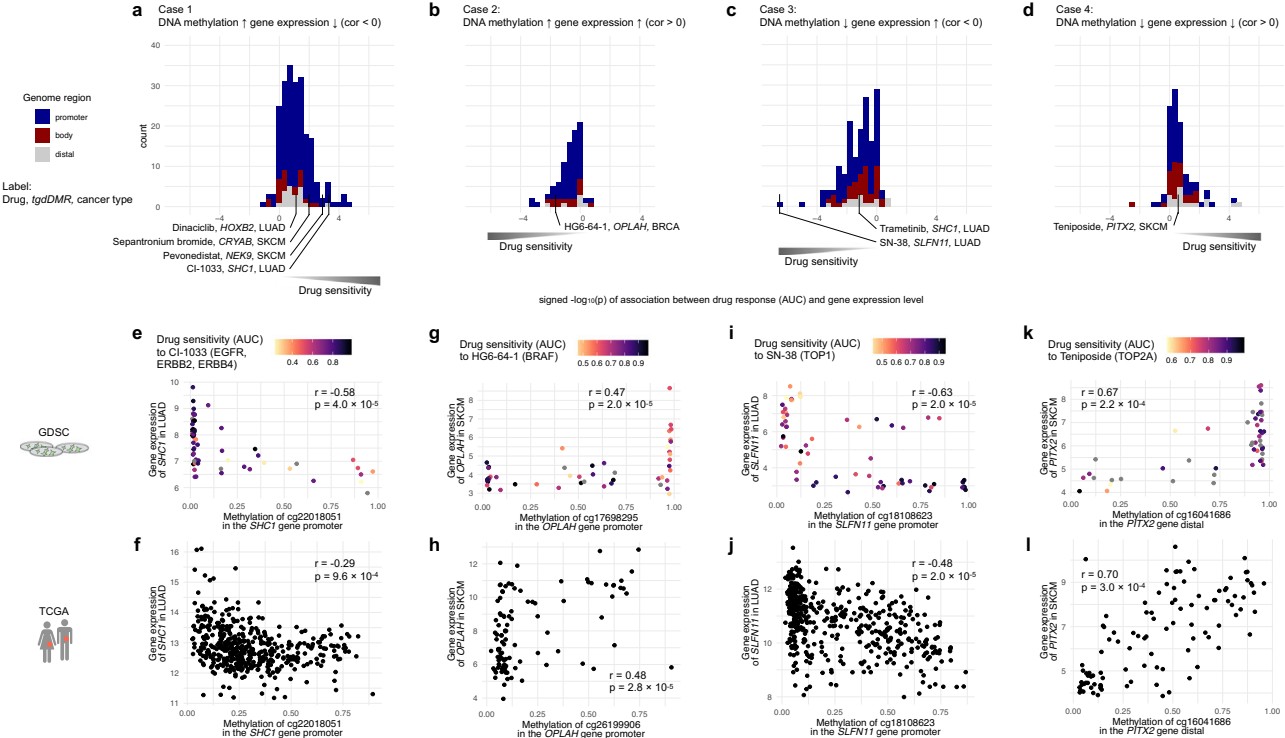

**Fig. 3 Epigenetic drug response biomarkers are empowered by studying DNA methylation and gene expression patterns.** This analysis revealed four distinct mechanisms observed across 377 dDMRs, i.e., Case 1-4: Cases 1 and 2 included dDMRs for which hypermethylation was associated with drug sensitivity and either **a** negative or **b** positive correlation with gene expression. For Cases 3 and 4 hypomethylated dDMRs were associated with drug sensitivity and either **c** negative or **d** positive correlation with gene expression. The x-axis shows the signed -$\log_{10}$(p-value) derived from a t-test of the coefficient of a linear model fit explaining drug response by proximal gene expression (Methods). Case 1 is exemplified by **e** the hypermethylation of the dDMR in the *SHC1* promoter regulating the expression in LUAD cancer cell lines, **f** which was validated in human tumour samples. In contrast, for Case 2 **g** hypermethylation in the *OPLAH* promoter promoted its expression in SKCM cell lines, and **h** tumour samples. For Case 3, **i** the hypermethylation of the *SLFN11* gene promoter downregulated the expression of *SFN11* in cancer cell lines, and **j** tumour samples. In Case 4, **k** positive correlations could be observed in the *PITX2* promoter and its expression in cell lines, and **l** tumour samples. The empirical adjusted p-value (p) for the respective CpG site and the Pearson correlation coefficient (r) are displayed. The used human icons are from the AIGA symbol signs collection and are in the public domain.

CCLE RRBS methylation data showed consistent effect sizes with an increased correlation of Pearson's $r = 0.85$ ($p = 0.01$, correlation test; Fig. 4c) compared to unfiltered dDMRs in the previous section. This highlights that reproducibility across independent drug screens and methylation datasets increased when focusing on tgdDMRs.

Currently, the majority of biomarkers for patient stratification are genetic alterations, thus, we investigated if genetic mutations and copy number alterations may reflect the methylation of tgdDMRs. We tested for associations between somatic mutations and tgdDMRs using linear models (Methods). We only observed weak correlations between somatic mutations and tgdDMRs (FDR < 0.1; Supplementary Fig. 7a; Methods).

While most tgdDMRs are found in gene promoters or bodies, we observed a distal region in a CpG island in the vicinity of the *HOXB2* gene that marked favourable drug responses for treatment with dinaciclib (CDK inhibitor), if the *HOXB2* tgdDMR was hypermethylated (dDMR calling, adj. $p < 10^{-6}$; Fig. 4d). Furthermore, the methylation status was correlated with *HOXB2* expression in cell lines (ELMER, emp. adj. $p < 0.001$; Fig. 4e) and primary tumours (ELMER, emp. adj. $p < 0.001$; Fig. 4f). Additionally, DNA repair enzyme encoding gene *APEX1* essentiality obtained from CRISPR knockout screens was significantly higher, if the tgdDMR was hypermethylated (FDR < 0.2; Supplementary Fig. 7d; Methods). HOX genes are a family of transcription factors that are frequently associated with cancer[45]. Their expression is reported to be regulated by DNA

methylation[46], however, the mechanisms by which they affect responses to dinaciclib remain elusive. Notably, we were able to validate this association in the independent CTRP drug screen (Pearson's $r = -0.59$, $p = 0.02$, correlation test; Supplementary Fig. 7b) and additionally observed consistent trends with an alternative methylation profiling based on RRBS in the CCLE (Pearson's $r = -0.48$, $p = 0.10$, correlation test; Supplementary Fig. 7c).

Next, we highlight further associations included in the identified tgdDMRs. For instance, hypermethylation of the tgdDMR in the *NEK9* promoter conferred sensitivity to NAE inhibition with pevonedistat in cell lines (dDMR calling, adj. $p < 10^{-6}$; Fig. 4g). In particular, we observed that tumours with hypermethylated tgdDMR in the *NEK9* promoter showed low *NEK9* expression in both cell lines (ELMER, emp. adj. $p < 0.001$; Fig. 4h) and patient tumours (ELMER, emp. adj. $p < 0.001$; Fig. 4i). NEK9 has been previously reported to participate in G1/S phase transition and progression and to regulate the kinase activity of CHK1 upon replication stress[47]. Examining the neighbourhood of signalling networks, the inhibition of NAE by pevonedistat leads to the inactivation of cullin-RING ligases[48], which target key proteins during the cell cycle progression such as CDK2 and CDC25A (Fig. 1h)[49]. This is supported by the Library of Integrated Network-Based Cellular Signatures (LINCS) database, which revealed the transcriptional dysregulation of *CUL3*, *CDC25A*, *CCNB1* and *PLK1* in SKCM cell lines upon treatment with pevonedistat (FDR < 0.1; Supplementary Fig. 7e;

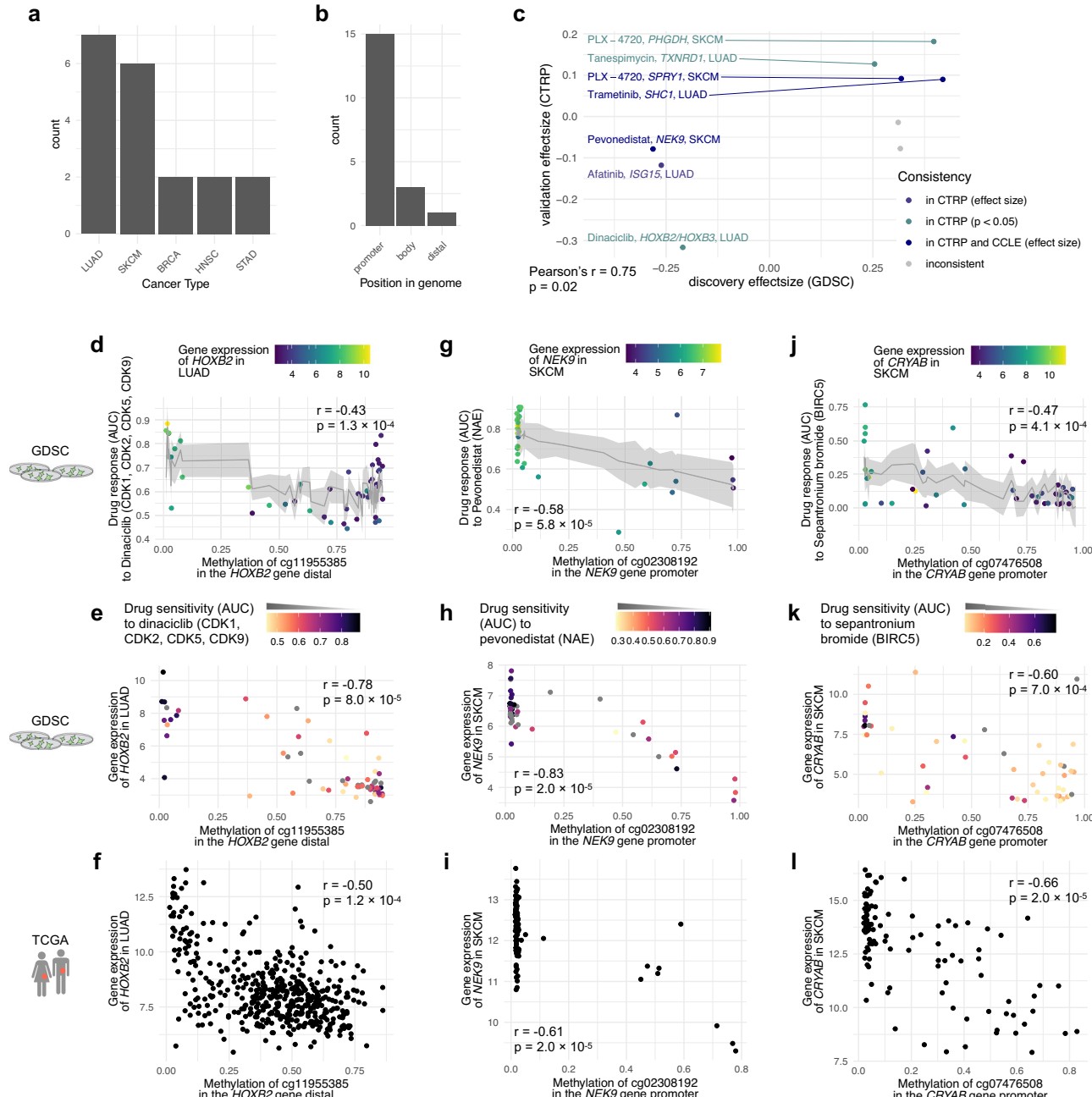

**Fig. 4 tgdDMRs in the context of lung cancer and melanoma. a** A histogram of tgdDMR in cancer types and **b** tgdDMR functional positions. **c** Scatter plot for validation of tgdDMRs, showing consistent effect sizes for CTRP and CCLE cohorts. DNA methylation of the distal dDMR in the vicinity of *HOXB2* is correlated with **d** response to dinaciclib and the expression of *HOXB2* in **e** cancer cell lines and **f** primary tumour samples. **g** Scatter plots show that the hypermethylated *NEK9* promoter confers sensitivity to pevonedistat in SKCM, the correlation between methylation in the *NEK9* promoter and its gene expression in SKCM in **h** cancer cell lines and **i** primary tumours. **j** Furthermore, scatter plots show that SKCM cell lines with hypomethylated *CRYAB* promoter do not respond to the apoptotic agent sepantronium bromide. Aberrant *CRYAB* expression with a hypermethylated promoter of *CRYAB* in **k** cell lines and **l** in tumour samples. For analysing DNA methylation and drug response, the error bars corresponding to 95% confidence intervals, the raw *p*-value (*p*) for each CpG site and the Pearson correlation coefficient (*r*) are reported. For analysing DNA methylation and gene expression, the empirical adjusted *p*-value (*p*) and the Pearson correlation coefficient (*r*) are reported. The used human icons are from the AIGA symbol signs collection and are in the public domain.

Methods). Concordantly, pevonedistat has been shown to induce DNA damage and cell cycle arrest[50,51], from which the cells with downregulated *NEK9* may not be able to recover.

A second tgdDMR in SKCM was identified, which involved a pro-apoptotic agent. Specifically, hypermethylation of the *CRYAB* promoter was associated with drug sensitivity to the BIRC5 inhibitor sepantronium bromide (dDMR calling, adj. $p < 10^{-6}$;

Fig. 4j) and aberrant *CRYAB* expression (ELMER, emp. adj. $p < 0.001$; Fig. 4k, l). Sepantronium bromide functions as a pro-apoptotic agent by inhibiting BIRC5, a member of the inhibitor of apoptosis (IAP) family[52]. The signalling network neighbourhood of the CRYAB tgdDMR shows interactions with CASP3 and P53 (Supplementary Fig. 5), which have been previously reported to show anti-apoptotic activity through CRYAB[53,54]. This

observation suggests that activated CRYAB may protect from apoptosis induced by sepantronium bromide, however, the exact nature of this relationship remains elusive. Nevertheless, the signalling network neighbourhoods of tgdDMRs offer interpretable indications about putative drug response mechanisms associated with tgdDMRs.

## Discussion

For advancing predictive epigenetic biomarkers in cancer, we presented an epigenome-wide multi-omic analysis for identifying interpretable and actionable epigenetic drug sensitivity biomarkers in HTS. In total, we identified 802 dDMRs demonstrating the epigenetic component of drug sensitivity in human cancer cell lines. Furthermore, we guided our method by the functional relationship that DNA methylation can mediate proximal gene expression, which resulted in a filtered set of 377 dDMRs that showed explainable regulation of transcriptional activity in human cancer cell lines. Furthermore, identifying consistency between cancer cell lines and primary tumours yielded evidence across epigenomic and transcriptomic data modalities and overcame limitations imposed by cell line artefacts[55]. This step prioritised 58 tgdDMRs of which 19 were further supported by protein-protein interaction networks. This thorough filtering was necessary because direct evidence of epigenetic biomarkers is lacking and validation was only possible for a limited number of dDMRs.

We observed an enrichment of cancer genes in the proximity of dDMRs, however, many established cancer genes lacked dDMRs, which suggests that only a minority of cancer genes may be epigenetically regulated. Furthermore, the modest correlations with somatic mutations suggest that DNA methylation may function complementary to genetic alterations for determining cancer drug susceptibilities. In contrast, DNA methylation was often accompanied by transcriptomic changes; however, it was not able to substitute DNA methylation pattern of dDMRs, i.e., more than half of dDMRs did not reveal regulations of a proximal gene. This suggests that tgdDMR methylation may either assist cancer cells in rewiring key signalling pathways through altering transcriptional signals or accompany other more elusive epigenetic mechanisms. This notion advocates our study design that first focuses on differentially methylated regions and consecutive integration of genetic and transcriptomic data. The layer-wise filtering starting with DNA methylation allowed us to evaluate intermediate results on all separate analysis steps and provide a comprehensive resource of epigenetic biomarkers (Supplementary Data 2).

Within this study, we focused on cancer type specific dDMRs and observed strong epigenetically diverse patterns across cancer types. Since the amount of found dDMRs was directly related to the studied sample size, we anticipate that forthcoming large-scale screening efforts can increase the power to detect dDMRs focusing on tumour subtypes, e.g., in BRCA[56] or COREAD[57]. Since DNA methylation can correlate with tumour subtypes, our analysis of dDMRs corrects for global methylation patterns through its principal components, which increases the ability to capture local mechanisms.

We showed consistency of tgdDMRs with an independent HTS and a different methylation profiling technology. Furthermore, we highlighted concordant epigenetic regulation of gene expression in human tumour samples, however, matched drug response readouts in human tumours are lacking. Nonetheless, our mechanisms may be validated in retrospective analyses of previously conducted molecularly characterised clinical trials for exploratory biomarker discovery. Although the signalling network neighbourhoods give insights into the potential mechanisms

for causal relationships or synthetically lethal interactions between drug targets and tgdDMRs-associated genes, tgdDMRs as predictive biomarkers remain to be further evaluated. In particular, melanoma patient subpopulations with promoter hypermethylation of tgdDMRs in the *NEK9* or *CRYAB* promoters could reveal benefits if treated with pevonedistat or pro-apoptotic agents such as sepantronium bromide, respectively.

We confirmed that DNA methylation in promoters is the major regulatory mechanism, and only sparse evidence supports mechanisms in gene bodies or distal regions. Thus, the role of methylation in cancer beyond its relevance in tumorigenesis and potential epigenetic vulnerabilities remains elusive. Upcoming technologies may enable the investigation of alternative epigenetic mechanisms in mediating drug responses beyond DNA methylation. For example, another class of epigenetic modifications, histone acetylation and histone methylation, are commonly associated with tumorigenesis and transcriptional regulations in cancer[58]. Furthermore, sequencing technologies beyond the traditional epigenome, e.g., ATAC-seq chromatin accessibility and Hi-C chromosome conformation, can yield further regulatory insights.

In essence, epigenetic data has the potential to yield the next generation of predictive biomarkers for precision medicine. The results of our analysis show that DNA methylation complemented with multi-omic data integration can reveal interpretable biomarkers for expanding the limited number of epigenetic biomarkers in clinical use. Our analysis for pharmaco-epigenomics can be applied to any drug screening effort with complementary multi-omics characterisation. Therefore, it may refine existing patient stratification and enhance the development of personalised cancer therapies in future.

## Methods

**Cancer cell lines and primary tumours**. We leveraged cancer cell lines from the Genomics of Drug Sensitivity in Cancer (GDSC) project[10] and the Cancer Cell Line Encyclopaedia (CCLE) project[12] as discovery and validation cohort, respectively. Both databases have been extensively characterised and curated[59]. The primary tumour samples are included in The Cancer Genome Atlas (TCGA), which aims to adhere to established guidelines and regulations regarding the use of human data[60]. Ethics and policies regarding the TCGA study are available at https://www.cancer.gov/about-nci/organization/ccg/research/structural-genomics/tcga/history/policies. Additional demographic characteristics of TCGA are available under https://portal.gdc.cancer.gov/ and have been reported previously[61].

**DNA methylation**. The raw methylation profiling data from GDSC, generated with the Infinium HumanMethylation450 BeadChip array, were downloaded from the Gene Expression Omnibus (GEO: accession number GSE68379 https://www.ncbi.nlm.nih.gov/geo/query/acc.cgi?acc=GSE68379). The data was processed with the R Bioconductor package Minfi[62], performing the noob background subtraction with dye-bias normalisation. After that, we filtered cross-reactive probes[63] and probes falling on sex chromosomes. The methylation beta-values were extracted and normalised by using the BMIQ method implemented in the R Bioconductor package ChAMP[64]. The probe annotations were obtained from the package IlluminaHumanMethylation450kanno.ilmn12.hg19[65].

The raw methylation profiling data from CCLE, generated with the reduced representation bisulfite sequencing (RRBS) methylation profiling technology, were downloaded in the form of fastq files from the Sequence Read Archive (SRA: accession number PRJNA523380 https://www.ncbi.nlm.nih.gov/bioproject/PRJNA523380/) using the SRA toolkit. We found 651 cell lines in our selected cancer types and performed quality control analysis and adaptor trimming using FastQC and TrimGalore[66], respectively. Subsequently, methylation percentage calls were retrieved from Bismark[67] using methylKit[68].

For the human primary tumours in TCGA, the preprocessed beta-values from the Infinium HumanMethylation450 BeadChip were downloaded from the GDC data portal (https://portal.gdc.cancer.gov/), accessed on the 18th October 2019. They were downloaded and processed with the R package TCGAbiolinks[69], using the ChAMP preprocessing pipeline consisting of filtering, imputation and normalisation methods with default parameters. Cancer types that either lacked DNA methylation or gene expression data, or had low sample size ($n < 8$), were excluded from further analysis, i.e., LAML, ALL, SCLC, NB, MM and OV.

**Gene expression**. For the cell lines in the GDSC project, we downloaded the RMA-processed Affymetrix array data from their website http://www.cancerrxgene.org /gdsc1000/, accessed on the 8th August 2019. For the human tumours, we downloaded the Hi-Seq count data from the RNAseq experiments in the TCGA database https://portal.gdc.cancer.gov/, accessed on the 18th October 2019. For the subsequent analysis, we performed variance stabilising transformation (VST) on the transcript count matrix.

**High-throughput drug response screens**. For the discovery cohort, we leveraged the HTS from the GDSC project http://www.cancerrxgene.org/downloads/bulk_download release 8.0. We limited the analysis to the 22 cancer types that had >15 fully treated and molecularly characterised cancer cell lines. Drug response was quantified by using the area-under-the-curve (AUC). A drug was required to display partial drug response across at least three cell lines, i.e., AUC ≤ 0.7. For the independent validation cohort, we used the Cancer Therapeutics Response Portal (CTRP) project https://portals.broadinstitute.org/ctrp.v2.1.

**Linear models and spatially correlated methylation sites for the identification of differentially methylated regions (dDMR calling)**. We employed a two-step analysis method to identify the differentially methylated regions of drug response (dDMRs). First, we identified differentially methylated sites in cancer cell lines. For that, we built linear models which fit the drug response denoted as $y$ by the methylation beta-value denoted as $m$ for each CpG site and drug in all cancer types, while correcting for the screening medium ($c_1$), growth properties ($c_2$), microsatellite instabilities ($c_3$) and the first two principal components ($pc_1, pc_2$) to correct for global methylation patterns. Thus, the linear model was defined by

$$y = \beta_0 + \beta_1 m + \beta_2 c_1 + \beta_3 c_2 + \beta_4 c_3 + \beta_5 pc_1 + \beta_6 pc_2, \qquad (1)$$

where $\beta_0, \ldots, \beta_6$ are the regression coefficients. The analysis was performed for each cancer type separately. The p-values were derived from the significance of the regression coefficient $\beta_1$ using a t-test for the respective CpG site. For the extraction of differentially methylated regions of drug response (dDMRs), we employed the software Comb-p[70,71] with default parameters. We first calculated the autocorrelation (ACF) between sites and the Stouffer-Liptak-Kechris correction of ACFs, followed by subsequent extraction of regions based on the Šidák-adjusted p-values (adj. $p$) while merging peaks within 1000 bases. dDMRs were called with a cutoff of adj. $p < 10^{-6}$. For the post-processing, the extracted regions were filtered such that there existed more than three cell lines that were aberrantly methylated for each dDMR. For this, we counted the number of cell lines which showed a methylation beta-value < 0.3 and beta-value > 0.7. Furthermore, we filtered regions for which the contained CpG sites did not meet the threshold for the raw $p < 0.01$. The identified region is labelled a dDMR, if both criteria were fulfilled. This subsequently yielded 802 drug differentially methylated regions (dDMRs) for 186 drugs. The effect size for each dDMR was defined as the mean of the regression coefficients $\beta_1$ across all CpG sites contained in the called region. The raw p-value ($p$) for each CpG site and the Pearson correlation coefficient ($r$) are reported for statistical tests analysing DNA methylation and drug response in the manuscript scatter plots.

**Inference of gene regulatory mechanisms as potential drug response biomarkers in cancer cell lines and human tumour samples**. To identify the proximal genes that were associated with aberrant methylation, we used the R package ELMER[39]. We focused on either promoter or distal regions within each cancer type[43]. For each dDMR, we tested the association between the methylation status and the gene expression with a Mann–Whitney U test according to the default parameters of ELMER[39]. We corrected for multiple hypothesis testing using a permutative approach with permutation size = 50000, raw p-value threshold = 0.05 and empirical adjusted p-value (emp. adj. $p$) threshold = 0.001. The empirical adjusted p-value ($p$) and the Pearson correlation coefficient ($r$) are reported for statistical tests analysing DNA methylation and gene expression in the manuscript scatter plots. In addition, for cancer cell lines, we tested if the proximal gene expression was associated with drug response independently of its dDMR. For this, we used linear models which fit the drug response to the respective proximal gene expression accordingly with the analogous linear models built using the methylation data.

**Protein-protein interaction networks between dDMR proximal genes and drug targets**. We identified protein-protein interaction networks in the neighbourhood of tgdDMR-associated genes and drug targets based on the OmniPath database[44]. For each of the 58 tgdDMRs, we extracted the correlated proximal gene and identified the ten shortest paths to each putative drug target using Yen's algorithm[72]. If no path from a gene to a drug target was found in the directed network, we identified paths traversing from the drug target to the tgdDMR gene. In summary, we were able to display protein-protein interaction networks with their shortest paths for 19/58 tgdDMRs, thus enhancing the mechanistic understanding of tgdDMRs.

**Somatic variants and their association with tgdDMRs**. The GDSC project has compiled a selection of somatic variants and copy number alterations[11], which are available at Cell Model Passports (https://cellmodelpassports.sanger.ac.uk/downloads). Only somatic mutations in coding regions were considered, which were binarised to represent the mutant and wild type status. Similarly, we binarised amplifications and deletions of gene-level copy number alterations. For both we only considered alterations which showed >3 altered cell lines. For assessing the correlation between genetic alterations and tgdDMRs, we used univariate linear models explaining tgdDMR methylation by the mutational status of each alteration. The p-values were derived from the significance of the regression coefficients and were multiplicity-adjusted by using the Benjamini–Hochberg method.

**CRISPR screens and their association with tgdDMRs**. CRISPR knockout data and associated gene effects on viability were downloaded from the DepMap Public 22Q4 primary files (https://depmap.org/portal/download/all/)[28,73]. Univariate linear models assessed associations between CRISPR knockouts for each gene in signalling network neighbourhoods of all tgdDMRs. The p-values were derived from the significance of the regression coefficients and were multiple hypothesis-adjusted by the Benjamini–Hochberg correction.

**LINCS drug transcriptomic signatures and their association with tgdDMRs**. We used the CLUE knowledge base (https://clue.io/lincs)[74] and its provided API to retrieve transcriptomic gene signatures from the overlapping compounds with matching tissue. Next, we tested for enrichments of each tgdDMR-associated gene and the corresponding genes in the signalling network neighbourhood in the set of gene signatures using a binomial test. The resulting p-values were adjusted using the Benjamini–Hochberg method.

**Statistics and reproducibility**. The sample sizes of the GDSC, CCLE/CTRP and TCGA data were predetermined by their data availability. We selected cancer types with >15 distinct molecularly characterised cell lines in the GDSC dataset. Cancer cell lines in the GDSC were parallelly treated according to the previously published study protocol[11]. For the matching cancer types, all distinct primary tumour samples with both available DNA methylation and gene expression data in the CCLE and TCGA data were selected. For all datasets, this resulted in 22 cancer types: small-cell lung cancer (SCLC; $n_{GDSC} = 63$; $n_{CCLE} = 36$; $n_{TCGA} = 0$), lung adenocarcinoma (LUAD; $n_{GDSC} = 63$; $n_{CCLE} = 87$; $n_{TCGA} = 484$), skin cutaneous melanoma (SKCM; $n_{GDSC} = 52$; $n_{CCLE} = 50$; $n_{TCGA} = 104$), breast invasive carcinoma (BRCA; $n_{GDSC} = 49$; $n_{CCLE} = 39$; $n_{TCGA} = 861$), colorectal adenocarcinoma (COREAD; $n_{GDSC} = 46$; $n_{CCLE} = 47$; $n_{TCGA} = 325$), head and neck squamous cell carcinoma (HNSC; $n_{GDSC} = 40$; $n_{CCLE} = 29$; $n_{TCGA} = 520$), glioblastoma (GBM; $n_{GDSC} = 35$; $n_{CCLE} = 37$; $n_{TCGA} = 51$), esophageal carcinoma (ESCA; $n_{GDSC} = 35$; $n_{CCLE} = 24$; $n_{TCGA} = 170$), ovarian serous cystadenocarcinoma (OV; $n_{GDSC} = 34$; $n_{CCLE} = 30$; $n_{TCGA} = 7$), lymphoid neoplasm diffuse large B-cell lymphoma (DLBC; $n_{GDSC} = 33$; $n_{CCLE} = 28$; $n_{TCGA} = 48$), neuroblastoma (NB; $n_{GDSC} = 32$; $n_{CCLE} = 14$; $n_{TCGA} = 0$), kidney renal clear cell carcinoma (KIRC; $n_{GDSC} = 30$; $n_{CCLE} = 21$; $n_{TCGA} = 344$), pancreatic adenocarcinoma (PAAD; $n_{GDSC} = 29$; $n_{CCLE} = 38$; $n_{TCGA} = 181$), acute myeloid leukemia (LAML; $n_{GDSC} = 25$; $n_{CCLE} = 29$; $n_{TCGA} = 0$), acute lymphocytic leukemia (ALL; $n_{GDSC} = 25$; $n_{CCLE} = 24$; $n_{TCGA} = 0$), stomach adenocarcinoma (STAD; $n_{GDSC} = 23$; $n_{CCLE} = 29$; $n_{TCGA} = 338$), mesothelioma (MESO; $n_{GDSC} = 21$; $n_{CCLE} = 8$; $n_{TCGA} = 86$), bladder urothelial carcinoma (BLCA; $n_{GDSC} = 19$; $n_{CCLE} = 24$; $n_{TCGA} = 428$), multiple myeloma (MM; $n_{GDSC} = 17$; $n_{CCLE} = 24$; $n_{TCGA} = 0$), liver hepatocellular carcinoma (LIHC; $n_{GDSC} = 17$; $n_{CCLE} = 20$; $n_{TCGA} = 412$), brain low-grade glioma (LGG; $n_{GDSC} = 17$; $n_{CCLE} = 15$; $n_{TCGA} = 511$) and thyroid carcinoma (THCA; $n_{GDSC} = 16$; $n_{CCLE} = 10$; $n_{TCGA} = 551$). The reproducibility of biomarkers was assessed by the overlapping CCLE/CTRP DNA methylation and drug response data as independent validation cohort. Discrepancies between drug response biomarkers in CCLE/CTRP may arise due to technical noise or differences in drug screening assays, but showed high consistency as reported.

**Reporting summary**. Further information on research design is available in the Nature Portfolio Reporting Summary linked to this article.

## Data availability

All datasets that were analysed in this study are publicly available within the outlined repositories. Specifically, the GDSC and CCLE DNA methylation data are available on Gene Expression Omnibus (GEO: accession number GSE68379) and Sequence Read Archive (SRA: accession number PRJNA523380), respectively. The TCGA DNA methylation data is available on the GDC data portal https://portal.gdc.cancer.gov/. The GDSC and CCLE drug response data are available on http://www.cancerrxgene.org/downloads/bulk_download release 8.0 and the Cancer Therapeutics Response Portal https://portals.broadinstitute.org/ctrp.v2.1, respectively. The GDSC and TCGA gene expression data are available on http://www.cancerrxgene.org /gdsc1000/ and the GDC data portal https://portal.gdc.cancer.gov/, respectively. The GDSC somatic variants and copy number alterations are available at Cell Model Passports https://cellmodelpassports.sanger.ac.uk/downloads. The CRISPR screens are available on DepMap https://depmap.

org/portal/download/all/ and the LINCS data is available on CLUE https://clue.io/lincs. The processed datasets are publicly available on Zenodo[75]. Source data for the figure panels are provided in Supplementary Data 3.

## Code availability

The source code for the presented analysis is available at https://github.com/MendenLab/pheb v0.1.0. It refers to a runnable docker image that contains all used software for data analysis. The statistical analysis can be reproduced with the source code and datasets provided on Zenodo[75].

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

## Acknowledgements
This project was supported by the European Union's Horizon 2020 Research and Innovation Programme (Grant agreement No. 950293-COMBAT-RES).

## Author contributions
Conceptualization: A.J.O. and M.P.M.; Data curation: A.J.O. and A.R.; Analysis, A.J.O. and M.P.M; Methodology: A.J.O. and M.P.M.; Supervision: M.P.M.; Visualisation: A.J.O.; Writing original draft: A.J.O. and M.P.M.; Writing, review and editing: A.J.O., A.R., G.A., G.K., E.G., D.S. and M.P.M.

## Funding

## Competing interests
M.P.M. collaborates with GSK, Roche and AstraZeneca, and receives funding from Roche and GSK. M.P.M. is a former employee at AstraZeneca. The remaining authors declare no competing interest.
