## [Peer Review File · Communications Biology]

The pharmacoepigenomic landscape of cancer cell lines reveals the epigenetic component of drug sensitivityReviewers' comments:

Reviewer #1 (Remarks to the Author):

The study analyzed 721 cancer cell lines across 22 cancer types and 453 anti-cancer compounds to identify DNA methylation biomarkers that predict treatment efficacy. The analysis detected 19 DNA methylation biomarkers across 17 drugs and five cancer types and found that DNA methylation mediates gene expression, enhancing biological signals across multi-omics data. The study discovered that NEK9 promoter hypermethylation may confer sensitivity to the NEDD8-activating enzyme (NAE) inhibitor pevonedistat in melanoma by downregulating NEK9. The study suggests that epigenomics will improve patient stratification in precision oncology.

The authors have performed substantial analysis to demonstrate DNA methylation as potential drug sensitivity biomarker.

However, I still have a questions about the results.

“The most prevalent cancer genes were APC and SKI found across two cancer types.”
APC is a cancer gene in colorectal, pancreatic, desmoid, hepatoblastoma, glioma, etc. Also, there are other critical cancer genes, such as KRAS, but these were not found associated with dDMRs. The author should confirm the results. If the cancer genes were less associated with dDMRs than non-cancer genes, the authors may raise a paragraph to discuss the findings.

Minor:

Grammar mistake:

The significance of HOX genes in cancer is frequency demonstrated and the regulation of their expression with DNA methylation was previously elucidated 42 , however, its mechanistic understanding is limited.

Reviewer #2 (Remarks to the Author):

In the manuscript «The pharmacoepigenomic landscape of cancer cell lines reveals the epigenetic component of drug sensitivity», Ohnmacht et al. use publicly available datasets to identify methylation sites with value as predictive biomarkers. They link the drug differentially methylated regions to gene expression in cell lines and human tumor samples, and identify functionally relevant biomarkers by connecting associated genes with corresponding drug targets through protein-protein signaling network analysis.

This is an interesting paper with what seems to me a solid analytical approach to identify functional relationships between multi-omics data and drug responses in cell lines. The paper identifies several specific omics-drug-relationships that can be analyzed and hopefully verified in (retro- or prospective) clinical studies. I believe the paper is of importance and interest to the scientific community. However, I have several comments to the article in the current form.

Major comments

1. While I understand that the main focus is identifying epigenetic mechanisms for drug-response, it would be interesting with some discussion around what the benefit of their approach is. To me (and I must admit that I am not an expert on epigenetics), it perhaps seems a little cumbersome, i.e. to first identify drug differentially methylated regions, then link/filter based on sites that are associated with gene expression, and then finally to filter based on genes that are connected with the drug target through protein signaling networks. Why not just link gene expression to drug response directly and

then filter based on protein signaling networks, and perhaps at the end of analysis link back to methylated events in the proximity of identified genes?

2. In the figures with linear fits, it would be nice with the actual fit, including r and p-values.

Minor comments

1. In results section, line 85: It says that the data from GDSC contained 453 anti-cancer compounds. When I looked at the paper they reference, it says that they tested 265 anti-cancer drugs. Is there an updated reference to refer to? Or where do the extra drugs tested come from?

2. It would be nice to state early in the results section that lower AUC scores indicate higher drug sensitivity. This is stated in the methods, and can be seen in figure 1b, but for me it would have been nice to just explicitly state this.

3. In the text for figure 1f: This sentence sounds strange to me: "In total, 58 epigenetic biomarkers of drug response were observed to be consistently methylated with the expression of their proximal gene in TCGA primary tumours". Perhaps "... were observed to be consistently methylated and associated with the expression...?"

4. In figure 1h: It is not obvious to me how NEK9 is connected within four steps to NAE. Can this be better highlighted in the figure? The shortest path I can easily see is through NAE-CUL3-RHOA-PAK1-PLK1-NEK9, but isn't this 5 steps?

5. In line 119: I'm not sure I would call variance between cell lines stemming from the same cancer type for intra-tissue variance. Perhaps change to "...highlighted that the variance within cancer types is lower compared to...".

6. In line 133: Perhaps change to "...and drugs that are involved in targeting mitosis...".

7. In line 147: Only 1 drug, change from "inhibitors" to "inhibitor".

8. In line 180: States that 227/1439 \diamond 28.3%. I guess the denominator should be 802.

9. In line 235: typo, OLPAH \diamond OPLAH.

10. In line 309: Do the authors mean "frequently demonstrated"?

11. In line 340: typo, EMLER \diamond ELMER

12. I'm not sure I follow the arguments in lines 337-344: Are the authors implying that CRYAB is involved in some sort of synthetic lethality mechanism for response?

13. In lines 439-443: Can this be reformulated, or better explained?

Specifically: "...denoted by γ by the methylation m -value denoted by m for less heteroscedasticity..." I'm not sure what is meant by "denoted by m for less heteroscedasticity".

14. In line 477: It says 19 tgdDMRs in main text, not 20.

15. There is a star next to Ohnmacht's name indicating equal contribution, but I do not see another star indicating who the other equal contributor is.

Reviewer #3 (Remarks to the Author):

Author remarks

The proposed paper titled The pharmacoepigenomic landscape of cancer cell lines reveals the epigenetic component of drug sensitivity discuss in details the relevance and intricacies of epigenetic alterations such as DNA methylation and how they can affect drug sensitivity in cancer cell lines. The approach proposed by Ohnmacht et al. is clearly discussed in detail, providing an in-depth investigation of DNA methylation and their connection with differential drug activity in a large sample of cancer cell lines. While the topic of the proposed paper is of actual high scientific relevance, some minor comments rose during revision of the manuscript, listed below.

Minor comments

- I have found the section Epigenetic biomarkers interpreted through gene regulatory mechanisms quite convoluted and confusing in the description of the cases, in relation to what correlates to what

(drug sensitivity and methylation levels, methylation levels and gene expression, all three together), specifically in the final discussion of the paragraph, from line 217 to line 239. On the other hand, the short text of Figure 3 describes the four cases in a much more simple and elegant way. I would suggest rephrasing the content of this paragraph to help the readers keep the focus on the straight correlation between methylation event and drug sensitivity (already assessed in the previous paragraph) while including another actor as gene expression, to avoid confusion.

- As discussed in the Discussion paragraph, the study relies on a limited set of data and could be improved by a wider pool of information to further investigate. On this note, the analysis focuses on the data provided by GDSC database. Are there any other database resources that could allow to further expand the data for the purpose of strengthening the conclusions discussed in the manuscript? Have the authors looked into the LINCS project and the vast array of data collected in that project?

- It seems very interesting that the methylation effect of the found tgDMRs does not correlate with any genetic alterations in the analysed cancer cell lines. This highlight, and the connected implications in terms of relevance of these epigenetic alterations, should be discussed more than a single small sentence. I would suggest the authors to further enhance this section, giving a new point of view of these alterations in term of patient stratification, as discussed in the later sections

**Reviewers' comments:**

**Referee expertise:**

**Referee #1: Cancer therapy, Pharmacogenomics**

**Referee #2: Genome Biology, Cancer Informatics**

**Referee #3: Computational Biology**

**Reviewers' comments:**

**Reviewer #1 (Remarks to the Author, cancer therapy, pharmacogenomics):**

**The study analyzed 721 cancer cell lines across 22 cancer types and 453 anti-cancer**
**compounds to identify DNA methylation biomarkers that predict treatment efficacy. The**
**analysis detected 19 DNA methylation biomarkers across 17 drugs and five cancer types**
**and found that DNA methylation mediates gene expression, enhancing biological signals**
**across multi-omics data. The study discovered that NEK9 promoter hypermethylation may**
**confer sensitivity to the NEDD8-activating enzyme (NAE) inhibitor pevonedistat in**
**melanoma by downregulating NEK9. The study suggests that epigenomics will improve**
**patient stratification in precision oncology.**

**The authors have performed substantial analysis to demonstrate DNA methylation as**
**potential drug sensitivity biomarker.**

We thank the reviewer for the positive feedback.

**However, I still have a questions about the results.**

**“The most prevalent cancer genes were APC and SKI found across two cancer types.”**
**APC is a cancer gene in colorectal, pancreatic, desmoid, hepatoblastoma, glioma, etc. Also,**
**there are other critical cancer genes, such as KRAS, but these were not found associated**
**with dDMRs. The author should confirm the results.**

Thank you for highlighting this. We indeed can confirm an enrichment of cancer genes in
dDMRs:

“[...] Furthermore, we investigated the dDMRs for regions in the proximity of cancer
 genes with the annotations from The Network of Cancer Genes (NCG) ²⁷, which were
 significantly enriched ($p = 0.01$; Fisher’s test).”

In addition, we evaluated possible biases of the 450k microarrays, i.e. investigated the seeding
 bias of methylation probes. We have revised our analysis accordingly as stated in **Results**
 and supplementary **Fig. S1e**:

“Furthermore, we investigated dDMRs in proximity of cancer genes based on
 annotations of the Network of Cancer Genes (NCG) project ²⁹. DNA Methylation sites
 on the 450k microarrays have higher seeding density in the vicinity of cancer genes,
 i.e. 645/674 (96%) of cancer genes contained > 10 profiled CpG sites compared to
 16,213/20,557 (79%) of non-cancer genes. To alleviate this bias, we only tested genes
 with at least ten proximal CpG sites, which resulted in 16,858 background genes and
 645/16,858 (3.8%) cancer genes. We observed 503 genes in proximity to identified
 dDMRs, of which 27 were cancer genes (5.4%; **Fig. S1e**), thus cancer genes were
 significantly enriched ($p = 0.049$; one-sided Fisher’s test).”

**Figure S1:** [...] (e) Set of 27 dDMRs proximal to cancer genes. [...]

**If the cancer genes were less associated with dDMRs than non-cancer genes, the authors**
**may raise a paragraph to discuss the findings.**

Cancer genes are enriched in dDMRs as described above, however, the fact that critical
oncogenes such as *KRAS* are not associated highlights that (i) only a few established cancer
genes are epigenetically altering drug susceptibilities; and (ii) DNA methylation likely functions
as a complementary mechanism to somatic mutations. We have expanded our **Discussion**
section as the following:

“[...] We observed an enrichment of cancer genes in the proximity of dDMRs, however,
many established cancer genes lacked dDMRs, which suggests that only a minority of
cancer genes may be epigenetically regulated. Furthermore, the modest correlations
with somatic mutations suggest that DNA methylation may function complementary to
genetic alterations for determining cancer drug susceptibilities. [...]”

In addition, we exemplify the established epigenetic regulation of *MGMT* in the context of BET
inhibitors now. Please see **Results** and **Figure S2f**:

“Among the cancer genes associated with dDMRs, we found that *MGMT* dDMR
methylation in low-grade glioma was associated with response to JQ1 (BET inhibitor;
**Fig. S1f**; $p = 3.7 \times 10^{-5}$, $r = -0.44$). The epigenetic silencing of *MGMT* is frequently
debated as a clinical biomarker³⁰ and previous work revealed that JQ1 disturbs DNA
damage responses by attenuating *MGMT* expression in glioblastoma cells³¹. While
the different treatment responses are often attributed to somatic mutations in cancer
genes, this suggests that DNA methylation can function as a complementary
mechanism.”

Figure S1: [...] (f) dDMR in *MGMT* for response to JQ1 in low-grade glioma (LGG).

**Minor:**

**Grammar mistake:**

**The significance of HOX genes in cancer is frequency demonstrated and the regulation of**
**their expression with DNA methylation was previously elucidated 42 , however, its**
**mechanistic understanding is limited.**

We thank the reviewer for pointing this out, we have revised this sentence as following:

“[...] HOX genes are a family of transcription factors that are frequently associated with
cancer⁴⁵. Their expression is reported to be regulated by DNA methylation⁴⁶, however,
the mechanisms by which they affect responses to dinaciclib remain elusive. [...]”

Furthermore, we double-checked the grammar throughout the whole manuscript and revised
it accordingly (track changes in blue). A selection of revised sentences is shown below:

“We selected cancer types with > 15 molecularly characterised cell lines in the GDSC
dataset. For all datasets, this resulted in 22 cancer types: [...]”

“[...] Thus, most molecular biomarker studies have focused on somatic mutations and
copy number variations. However, despite [...]”

“We employed a two-step analysis method to identify the differentially methylated
regions of drug response (dDMRs). [...]”

All grammar corrections are thoroughly highlighted throughout the manuscript.

**Reviewer #2 (Remarks to the Author, Genome Biology, Cancer Informatics):**

**In the manuscript «The pharmacoepigenomic landscape of cancer cell lines reveals the**
**epigenetic component of drug sensitivity», Ohnmacht et al. use publicly available datasets**
**to identify methylation sites with value as predictive biomarkers. They link the drug**
**differentially methylated regions to gene expression in cell lines and human tumor samples,**
**and identify functionally relevant biomarkers by connecting associated genes with**
**corresponding drug targets through protein-protein signaling network analysis.**

**This is an interesting paper with what seems to me a solid analytical approach to identify**
**functional relationships between multi-omics data and drug responses in cell lines. The**
**paper identifies several specific omics-drug-relationships that can be analyzed and**
**hopefully verified in (retro- or prospective) clinical studies. I believe the paper is of**
**importance and interest to the scientific community.**

We thank the reviewer for his/her kind feedback.

**However, I have several comments to the article in the current form.**

**Major comments**

**1. While I understand that the main focus is identifying epigenetic mechanisms for drug-**
**response, it would be interesting with some discussion around what the benefit of their**
**approach is. To me (and I must admit that I am not an expert on epigenetics), it perhaps**
**seems a little cumbersome, i.e. to first identify drug differentially methylated regions, then**
**link/filter based on sites that are associated with gene expression, and then finally to filter**
**based on genes that are connected with the drug target through protein signaling networks.**
**Why not just link gene expression to drug response directly and then filter based on protein**
**signaling networks, and perhaps at the end of analysis link back to methylated events in**
**the proximity of identified genes?**

Thanks for this valuable comment. It is fair that other study designs may yield similar final
results, however, the strength of our design is the focus on epigenetic mechanisms in cancer.
Notably, gene expression may be regulated by epigenetic mechanisms, but there are also
only imperfect correlations. Starting with drug differentially methylated regions and
consecutively layer-wise filtering and reporting all intermediate results empowers to leverage
these in follow-up studies and provides a comprehensive epigenetic biomarker resource
(Table S2). We added these points to our Discussion:

“[...] This notion advocates our study design that first focuses on differentially
methylated regions and consecutive integration of genetic and transcriptomic data.
The layer-wise filtering starting with DNA methylation allowed us to evaluate
intermediate results on all separate analysis steps and provide a comprehensive
resource of epigenetic biomarkers (Table S2). [...]”

If we first analysed gene expression and consecutively filtered for concordant epigenetic links,
important gold standard epigenetic associations would be missed. For example, the known
epigenetic regulation of the *MGMT* promoter was detected by our analysis, but did not show
significant associations with its expression (Fig. S1f). Please see the added example based
on an intermediate analysis step:

“Among the cancer genes associated with dDMRs, we found that *MGMT* dDMR
methylation in low-grade glioma was associated with response to JQ1 (BET inhibitor;
Fig. S1f; $p = 3.7 \times 10^{-5}$, $r = -0.44$). The epigenetic silencing of *MGMT* is frequently
debated as a clinical biomarker³⁰ and previous work revealed that JQ1 disturbs DNA
damage responses by attenuating *MGMT* expression in glioblastoma cells³¹. While
the different treatment responses are often attributed to somatic mutations in cancer
genes, this suggests that DNA methylation can function as a complementary
mechanism.”

**Figure S1:** [...] (f) dDMR in *MGMT* for response to JQ1 in low-grade glioma (LGG).

Furthermore, this raises interesting hypotheses how DNA methylation may function as a
complementary oncogenic mechanism, which we added to the **Discussion**:

“[...] In contrast, DNA methylation was often accompanied by transcriptomic changes;
however, it was not able to substitute DNA methylation pattern of dDMRs, i.e. more
than half of dDMRs did not reveal regulations of a proximal gene. This suggests that
tgDMMR methylation may either assist cancer cells in rewiring key signalling pathways

through altering transcriptional signals or accompany other more elusive epigenetic
mechanisms. [...]”

**2. In the figures with linear fits, it would be nice with the actual fit, including r and p-values.**

Thank you for raising this comment. In the revised version, we have added r-values and p-
values for all statistical tests. We acknowledge this also in the **Methods** section:

“[...] The raw p-value (p) for each CpG site and the Pearson correlation coefficient (r)
are reported for statistical tests analysing DNA methylation and drug response.”

“[...] The empirical adjusted p-value (p) and the Pearson correlation coefficient (r) are
reported for statistical tests analysing DNA methylation and gene expression. [...]”

Specifically, for the scatter plots that show drug response and DNA methylation, we now
include the original r-values and p-values from the linear model fits which were used as input
to the comb-p algorithm to find dDMRs. Furthermore, we removed the regression lines from
the scatter plots that show DNA methylation and gene expression because nonparametric
Mann-Whitney U tests were used and added the r-values and p-values accordingly. The
revised figures are shown below:

**Figure 2:** [...] (e) The association between *SHC1* promoter hypermethylation and CI-1033 response in
LUAD; and (f) the association between *SLFN11* gene hypomethylation and response to SN-38 with the
raw p-value (p) for the respective CpG site and the Pearson correlation coefficient (r).

Figure 3: [...] The x-axis gene shows the signed $-\log_{10}(p\text{-value})$ of a univariate linear model explaining drug response by proximal gene expression (**Methods**). Case 1 is exemplified by (e) the hypermethylation of the dDMR in the *SHC1* promoter regulating the expression in LUAD cancer cell lines, (f) which was validated in human tumour samples. In contrast, for Case 2 (g) hypermethylation in the *OPLAH* promoter promoted its expression in SKCM cell lines, and (h) tumour samples. For Case 3, (i) the hypermethylation of the *SLFN11* gene promoter downregulated the expression of *SFN11* in cancer cell lines, and (j) tumour samples. In Case 4, (k) positive correlations could be observed in the *PITX2* promoter and its expression in cell lines, and (l) tumour samples.

Figure 4: [...] (d) The expression of the tgdDMR for dinaciclib in the vicinity of *HOXB2* is correlated with tgdDMR methylation in (e) cancer cell lines and (f) primary tumour samples. (g) Scatter plots show that the hypermethylated *NEK9* promoter confers sensitivity to pevonedistat in SKCM, the correlation between methylation in the *NEK9* promoter and its gene expression in SKCM in (h) cancer cell lines and (i) primary tumours. (j) Furthermore, scatter plots show that SKCM cell lines with hypomethylated *CRYAB* promoter do not respond to the apoptotic agent sepantronium bromide. Aberrant *CRYAB* expression with a hypermethylated promoter of *CRYAB* in (k) cell lines and (l) in tumour samples.

Minor comments

1. In results section, line 85: It says that the data from GDSC contained 453 anti-cancer compounds. When I looked at the paper they reference, it says that they tested 265 anti-cancer drugs. Is there an updated reference to refer to? Or where do the extra drugs tested come from?

We appreciate this comment and indeed the original publication from 2016 only contained 265 screened drugs, which is referred to as GDSC1 release according to their website. The GDSC2 data release was included in some recent publications (Picco et al. 2019 and Gonçalves et al. 2020)^{27,28}. This release almost doubled the amount of screened compounds to a total of 453 unique compounds, which we included in our analysis.

We now acknowledge the increased amount of screened compounds in the **Results**:

“For the discovery of DNA methylation biomarkers of drug response, we analysed
methylation patterns of 721 cancer cell lines from 22 cancer types treated with 453
anti-cancer compounds. The data was derived from the Genomics of Drug Sensitivity
in Cancer (GDSC; **Fig. 1a**) project ¹¹, which has since expanded its set of screened
compounds compared to the original publication ^{27,28}. [...]”

Also, we added in the Method section the download links of used data versions:

“For the discovery cohort, we leveraged high-throughput drug screens from the GDSC
project http://www.cancerrxgene.org/downloads/bulk_download release 8.0.”

**2. It would be nice to state early in the results section that lower AUC scores indicate higher
drug sensitivity. This is stated in the methods, and can be seen in figure 1b, but for me it
would have been nice to just explicitly state this.**

Thank you for pointing this out, we have added this definition also in the beginning of the
**Results** section upon its first introduction:

“[...] Drug responses of cancer cell lines were characterised by their area under the
drug response curve (AUC; **Fig. 1b**), for which low AUC values convey high sensitivity
to the respective compound.”

**3. In the text for figure 1f: This sentence sounds strange to me: “In total, 58 epigenetic
biomarkers of drug response were observed to be consistently methylated with the
expression of their proximal gene in TCGA primary tumours”. Perhaps “... were observed
to be consistently methylated and associated with the expression...”?**

We would like to thank the reviewer for highlighting this flawed sentence, which we corrected
as the following:

**Figure 1:** [...] (f) In total, the methylation of 58 epigenetic biomarkers of drug response were observed to
be consistently correlated with the expression of their proximal gene in TCGA primary tumours. [...]”

Furthermore, we thoroughly reviewed the grammar and revised the language throughout the
whole manuscript. A selected set of revised sentences is shown in the following examples:

“We selected cancer types with > 15 molecularly characterised cell lines in the GDSC
dataset. For all datasets, this resulted in 22 cancer types: [...]”

“[...] Thus, most molecular biomarker studies have focused on somatic mutations and
copy number variations. However, despite [...]”

“We employed a two-step analysis method to identify the differentially methylated
regions of drug response (dDMRs). [...]”

All edits are highlighted in the manuscript.

**4. In figure 1h: It is not obvious to me how NEK9 is connected within four steps to NAE. Can
this be better highlighted in the figure? The shortest path I can easily see is through NAE-
CUL3-RHOA-PAK1-PLK1-NEK9, but isn't this 5 steps?**

Thank you a lot and we apologise for this mistake. It is indeed five steps, which was corrected
accordingly. Furthermore, we agree that the shortest paths in the shown networks were not
immediately visible, which may lead to misunderstandings. Thus, we now highlight the shortest
path by colour code, which is acknowledged in the figure legends:

**Figure 1:** [...] (h) The predictive biomarker *NEK9* (light-blue) is connected within five steps to the drug
target of pevonedistat, i.e., the NEDD8-activating enzyme NAE (pink). In the graph, nodes that are
traversed with a shortest path are highlighted by the blue-grey colour among the alternative paths.

Figure S4: tgdDMRs in LUAD. (a)-(ab) Correlation between drug response quantified by area-under-the-curve, DNA methylation and gene expression plus the corresponding protein-protein interaction network between putative drug target (pink) and tgdDMR-associated gene encoding protein (light-blue). In the graph, nodes that are traversed with a shortest path are highlighted by the blue-grey colour among the alternative paths.

Figure S5: tgdDMRs in SKCM. (a)-(w) Correlation between drug response quantified by area-under-the-curve, DNA methylation and gene expression plus the corresponding protein-protein interaction network between putative drug target (pink) and tgdDMR-associated gene encoding protein (light-blue). In the graph, nodes that are traversed with a shortest path are highlighted by the blue-grey colour among the alternative paths.

Figure S6: tgdDMRs in BRCA, HNSC and STAD. (a)-(w) Correlation between drug response quantified by area-under-the-curve, DNA methylation and gene expression plus the corresponding protein-protein interaction network between putative drug target (pink) and tgdDMR-associated gene encoding protein (light-blue). In the graph, nodes that are traversed with a shortest path are highlighted by the blue-grey colour among the alternative paths.

**5. In line 119: I'm not sure I would call variance between cell lines stemming from the same**
 **cancer type for intra-tissue variance. Perhaps change to "...highlighted that the variance**
 **within cancer types is lower compared to..."**

Thank you for the comment, we agree and have adjusted this sentence accordingly:

"Analysing the DNA methylation and gene expression profiles of cancer cell lines
 stemming from 22 cancer types highlighted that the variance within cancer types is
 lower compared to the variance between cancer types (Fig. 2a; Fig. S1a). [...]"

**6. In line 133: Perhaps change to "...and drugs that are involved in targeting mitosis..."**

Thank you, we also revised this according to the reviewer's suggestion.

**7. In line 147: Only 1 drug, change from “inhibitors” to “inhibitor”.**

We thank the reviewer for identifying this typo, which is corrected in the revised manuscript.

**8. In line 180: States that 227/1439 28.3%. I guess the denominator should be 802.**

We apologise for this transcription error. Indeed the reviewer is right, the denominator is
changed to 802 in the revised version.

**9. In line 235: typo, OLPAH OPLAH.**

We apologise for this typo and have corrected it accordingly. We also systematically double-
checked all the used gene names and abbreviations for typos, and revised if necessary.

**10. In line 309: Do the authors mean “frequently demonstrated”?**

Since this grammar mistake was also raised by reviewer #1, we decided to rephrase this
sentence accordingly. It now reads:

“[...] HOX genes are a family of transcription factors that are frequently associated with
cancer⁴⁵. Their expression is reported to be regulated by DNA methylation⁴⁶, however,
the mechanisms by which they affect responses to dinaciclib remain elusive. [...]”

**11. In line 340: typo, EMLER ELMER**

Similar as above, this mistake prompted us to carefully double-check the whole manuscript
for typos in gene names and abbreviations, which we corrected accordingly.

**12. I'm not sure I follow the arguments in lines 337-344: Are the authors implying that**
**CRYAB is involved in some sort of synthetic lethality mechanism for response?**

The concept of synthetically lethal interactions implies a causal relationship between two
proteins, i.e. CRYAB and BIRC5, which we cannot claim since our reported epigenetic
biomarkers are purely associative. However, the signalling neighbourhood of tgdDMRs can
give valuable insights into potential mechanisms, which yields another level of evidence of the

*tgdDMRs* that were found in a purely data-driven way. In conjunction with the previous
comment (point 4.), the supplied colour scheme supports the interpretation of a potential
mechanism for *CRYAB* in the network in **Fig. S5**:

Figure S5: *tgdDMRs* in SKCM. (a)-(w) Correlation between drug response quantified by area-under-the-curve, DNA methylation and gene expression plus the corresponding protein-protein interaction network between putative drug target (pink) and *tgdDMR*-associated gene encoding protein (light-blue). In the graph, nodes that are traversed with the shortest path are highlighted by the pink-grey colour and nodes in the other alternative shortest paths are coloured in blue-grey.

Consequently, we also revised the respective **Results** section:

“A second *tgdDMR* in SKCM was identified, which involved a pro-apoptotic agent.
Specifically, hypermethylation of the *CRYAB* promoter was associated with drug
sensitivity to the BIRC5 inhibitor sepantronium bromide (dDMR calling, adj. $p < 10^{-6}$,
**Fig. 5j**) and aberrant *CRYAB* expression (*ELMER*, FDR < 0.001 , **Fig. 4k,l**).
Sepantronium bromide functions as a pro-apoptotic agent by inhibiting BIRC5, a
member of the inhibitor of apoptosis (IAP) family⁵². The signalling network
neighbourhood of the *CRYAB* *tgdDMR* shows interactions with CASP3 and P53 (**Fig.**
**S5o**), which have been previously reported to show anti-apoptotic activity through
*CRYAB*^{53,54}. This observation suggests that activated *CRYAB* may protect from
apoptosis induced by sepantronium bromide, however, the exact nature of this
relationship remains elusive. Nevertheless, the signalling network neighbourhoods of
*tgdDMRs* offer interpretable indications about putative drug response mechanisms
associated with *tgdDMRs*.”

Furthermore, we also revised respective paragraph in the **Discussion** section to make clear
the limitations of our approach:

“We showed consistency of *tgdDMRs* with an independent high-throughput drug
screen and a different methylation profiling technology. Furthermore, we highlighted

concordant epigenetic regulation of gene expression in human tumour samples,
however, matched drug response readouts in human tumours are lacking.
Nonetheless, our mechanisms may be validated in retrospective analyses of
previously conducted molecularly characterised clinical trials for exploratory biomarker
discovery. Although the signalling network neighbourhoods give insights into the
potential mechanisms for causal relationships or synthetically lethal interactions
between drug targets and tgdDMRs-associated genes, tgdDMRs as predictive
biomarkers remain to be further evaluated. In particular, melanoma patient
subpopulations with promoter hypermethylation of tgdDMRs in the *NEK9* or *CRYAB*
promoters could reveal benefits if treated with pevonedistat or pro-apoptotic agents
such as sepantronium bromide, respectively.”

**13. In lines 439-443: Can this be reformulated, or better explained?**

**Specifically: “...denoted by y by the methylation m -value denoted by m for less**
**heteroscedasticity...” I’m not sure what is meant by “denoted by m for less**
**heteroscedasticity”.**

We apologise for the confusion. We decided to consistently use beta-values across the whole
manuscript since they give consistent results and do not face the heteroscedasticity issue.
Thus, we simply removed this sentence from the manuscript.

**14. In line 477: It says 19 tgdDMRs in main text, not 20.**

Thanks, this typo has been corrected.

**15. There is a star next to Ohnmacht’s name indicating equal contribution, but I do not see**
**another star indicating who the other equal contributor is.**

Thanks, we removed the asterisk.

**Reviewer #3 (Remarks to the Author, Computational Biology):**

**Author remarks**

**The proposed paper titled The pharmacoepigenomic landscape of cancer cell lines reveals**
**the epigenetic component of drug sensitivity discuss in details the relevance and**
**intricacies of epigenetic alterations such as DNA methylation and how they can affect drug**
**sensitivity in cancer cell lines. The approach proposed by Ohnmacht et al. is clearly**
**discussed in detail, providing an in-depth investigation of DNA methylation and their**
**connection with differential drug activity in a large sample of cancer cell lines. While the**
**topic of the proposed paper is of actual high scientific relevance, some minor comments**
**rose during revision of the manuscript, listed below.**

We thank the reviewer for the positive feedback.

**Minor comments**

**- I have found the section Epigenetic biomarkers interpreted through gene regulatory**
**mechanisms quite convoluted and confusing in the description of the cases, in relation to**
**what correlates to what (drug sensitivity and methylation levels, methylation levels and**
**gene expression, all three together), specifically in the final discussion of the paragraph,**
**form line 217 to line 239. On the other hand, the short text of Figure 3 describes the four**
**cases in a much more simple and elegant way. I would suggest rephrasing the content of**
**this paragraph to help the readers keep the focus on the straight correlation between**
**methylation event and drug sensitivity (already assessed in the previous paragraph) while**
**including another actor as gene expression, to avoid confusion.**

Thank you for this comment. We agree that the description of the cases would benefit from
clarification. In order to address this, we decided to reorder the columns in **Fig. 3** matching
the order of the manuscript.

Figure 3: Epigenetic drug response biomarkers are empowered by studying DNA methylation and gene expression patterns. This analysis revealed four distinct mechanisms observed across 377 dDMRs, i.e. *Cases 1-4*: *Cases 1 and 2* included dDMRs for which hypermethylation was associated with drug sensitivity and either (a) negative or (b) positive correlation with gene expression. For *Cases 3 and 4* hypomethylated dDMRs were associated with drug sensitivity and either (c) negative or (d) positive correlation with gene expression. The x-axis shows the signed $-\log_{10}(p\text{-value})$ of a univariate linear model explaining drug response by proximal gene expression (**Methods**). *Case 1* is exemplified by (e) the hypermethylation of the dDMR in the *SHC1* promoter regulating the expression in LUAD cancer cell lines, (f) which was validated in human tumour samples. In contrast, for *Case 2* (g) hypermethylation in the *OPLAH* promoter promoted its expression in SKCM cell lines, and (h) tumour samples. For *Case 3*, (i) the hypermethylation of the *SLFN11* gene promoter downregulated the expression of *SFN11* in cancer cell lines, and (j) tumour samples. In *Case 4*, (k) positive correlations could be observed in the *PITX2* promoter and its expression in cell lines, and (l) tumour samples.

Furthermore, we rephrased the paragraphs of this section for a smoother introduction to the concepts while focusing on drug responses:

“[...] In summary, we observed four distinct mechanisms which may drive drug sensitivity, i.e., hypermethylation with either downregulated gene expression (*Case 1*, $n = 216$; **Fig. 3a**) or upregulated gene expression (*Case 2*, $n = 110$; **Fig. 3b**), and hypomethylation with either upregulated gene expression (*Case 3*, $n = 162$; **Fig. 3c**) or downregulated gene expression (*Case 4*, $n = 88$; **Fig. 3d**). We exemplified each case in cancer cell lines and their mechanistic consistency in primary tumours (**Fig. 3e-l**).

For both Cases 1 and 2, hypermethylated dDMRs were associated with drug sensitivity
(negative effect size in Fig. 2b). The majority of dDMRs belonged to Case 1, which
was distinguished by promoter regions (Fig. 3a). It resembles the canonical
mechanism in which hypermethylation of promoter regions downregulates the
expression of their associated proximal gene and thereby confers drug sensitivity. This
behaviour is exemplified by the methylation of the *SHC1* promoter and its gene
expression in LUAD cell lines (Fig. 3e). Additionally, we verified the association of the
epigenetic status and gene expression in LUAD human tumour samples (Fig. 3f).

For Case 2, hypermethylation of dDMRs correlated with higher expression of proximal
genes (Fig. 3g,h). This is a less frequent epigenetic regulation mechanism, however,
it is consistent with previous studies reporting both behaviours^{8,40–42}. As an example,
the hypermethylation of the *OPLAH* dDMR was associated with the upregulation of
*OPLAH* expression in SKCM cancer cell lines and HG-6-64-1 drug sensitivity (Fig. 3g).
In addition, this epigenetic regulation of *OPLAH* expression was also demonstrated in
primary tumour samples (Fig. 3h).

Cases 3 and 4 were characterised by hypomethylated dDMRs that were associated
with drug sensitivity (positive effect size in Fig. 2b), which could also be distinct by
negative or positive correlations of dDMRs with gene expression for Case 3 and Case
4, respectively. For example, we found that the hypomethylation of the *SLFN11* dDMR
in LUAD was associated with higher *SLFN11* expression (Fig. 3i), which was further
verified in human tumour samples (Fig. 3j). In contrast, the hypomethylation of *PITX2*
dDMR was linked to teniposide drug sensitivity, however, the hypermethylation of
*PITX2* dDMR was positively associated with *PITX2* expression in cancer cell lines and
human tumour samples (Fig. 3k,l).

In summary, drug sensitivity in cancer cell lines may be driven by either
hypermethylation (Cases 1 and 2) or hypomethylation (Cases 3 and 4) of dDMRs and
can either present negatively correlated gene expression (Cases 1 and 3) or positively
correlated gene expression (Cases 2 and 4). Case 1 has been the focus of most
epigenetic biomarker studies, whilst we systematically investigated all 4 cases (Table
S2) and therefore can provide broader mechanistic insights.”

- As discussed in the Discussion paragraph, the study relies on a limited set of data and
could be improved by a wider pool of information to further investigate. On this note, the
analysis focuses on the data provided by GDSC database. Are there any other database

resources that could allow to further expand the data for the purpose of strengthening the
conclusions discussed in the manuscript?

The reviewer raises an important point. We successfully incorporated additional tumour data
from the TCGA and drug response data with matched molecular profiles from the CTRP and
CCLE, which was able to support the conclusions of the set of tgDMDRs. In particular, we
believe that further phenotypic readouts in TCGA would strengthen our conclusions.
Unfortunately, further human clinical data is lacking in the public domain, which is
acknowledged in the manuscript:

“[...] Furthermore, we highlighted concordant epigenetic regulation of gene expression
in human tumour samples, however, matched drug response readouts in human
tumours are lacking. [...]”

We further expanded our analysis including CRISPR knockout screens in cancer cell lines.
This allowed us to gain additional insights into the regulated pathway neighbourhood of
tgDMDR-associated genes to alter cell viability. In essence, we conducted an analysis to test
for associations between CRISPR knockouts of tgDMDRs and their genes of identified
pathway neighbourhoods using linear regression shown in **Fig. S7d**.

**Figure S7:** [...] (d) A volcano plot summarising associations between tgDMDRs and CRISPR knockout
screens of the genes associated with tgDMDRs and their signalling network neighbourhood. [...]

Notably, the previously highlighted tgDMDR of dinaciclib is further supported by this analysis,
thus we included this in the **Results**:

“[...] While most tgDMMRs are found in gene promoters or bodies, we observed a distal
region in a CpG island in the vicinity of the *HOXB2* gene that marked favourable drug
responses for treatment with dinaciclib (CDK inhibitor), if the *HOXB2* tgDMMR was
hypermethylated (dDMR calling, adj. $p < 10^{-6}$, **Fig. 4d**). Furthermore, the methylation
status was correlated with *HOXB2* expression in cell lines (**Fig. 4e**) and primary
tumours (**Fig. 4f**). Additionally, DNA repair enzyme encoding gene *APEX1* essentiality
obtained from CRISPR knockout screens was significantly higher, if the tgDMMR was
hypermethylated (FDR < 0.2; **Fig. S7d**; **Methods**). [...]”

We also expanded the **Methods** accordingly:

“**CRISPR screens and their association with tgDMMRs**

CRISPR knockout data and associated gene effects on viability were downloaded from
the DepMap Public 22Q4 primary files (<https://depmap.org/portal/download/all/>)^{68,69}.
Univariate linear models assessed associations between CRISPR knockouts for each
gene in signalling network neighbourhoods of all tgDMMR. The p-values were derived
from the significance of the regression coefficients and were multiple hypothesis-
adjusted by the Benjamini-Hochberg correction.”

**Have the authors looked into the LINCS project and the vast array of data collected in that**
**project?**

Thank you for the suggestion. For overlapping compounds and screened cancer types, we
have tested for enrichment of genes associated to tgDMMR and their signalling network
neighbourhood in the LINCS drug signatures. We added the results for this assessment to
**Fig. S7e:**

Figure S7: [...] (e) A volcano plot summarising enrichments of genes associated with tgDMMRs and their signalling network neighbourhood in the LINCS drug signatures for the matching compound and cancer type.

Accordingly, we were able to identify that transcriptional dysregulation of *CUL3*, *CDC25A*, *CCNB1* and *PLK1* upon treatment of SKCM cell lines with pevonedistat, which we added in the respective paragraph in the **Results** section:

“[...] Examining the neighbourhood of signalling networks, the inhibition of NAE by pevonedistat leads to the inactivation of cullin-RING ligases⁴⁸, which target key proteins during the cell cycle progression such as CDK2 and CDC25A (**Fig. 1h**)⁴⁹. This is supported by the Library of Integrated Network-Based Cellular Signatures (LINCS) database, which revealed the transcriptional dysregulation of *CUL3*, *CDC25A*, *CCNB1* and *PLK1* in SKCM cell lines upon treatment with pevonedistat (FDR < 0.1; **Fig. S7e; Methods**). Concordantly, pevonedistat has been shown to induce DNA damage and cell cycle arrest^{50,51}, from which the cells with downregulated *NEK9* may not be able to recover. [...]”

Additionally, we also added the results of the additional data sources as annotations in the tgDMMR summary table in **Table S2**. We also added the details of the analysis to the **Methods** section:

“LINCS drug transcriptomic signatures and their association with tgDMMRs

We used the CLUE knowledge base (<https://clue.io/lincs>)⁷¹ and its provided API to
 retrieve transcriptomic gene signatures from the overlapping compounds with
 matching tissue. Next, we tested for enrichments of each tgDMMR-associated gene
 and the corresponding genes in the signalling network neighbourhood in the set of
 gene signatures using a binomial test. The resulting p-values were adjusted using the
 Benjamini-Hochberg method.”

 - It seems very interesting that the methylation effect of the found tgDMMRs does not
 correlate with any genetic alterations in the analysed cancer cell lines. This highlight, and
 the connected implications in terms of relevance of these epigenetic alterations, should be
 discussed more than a single small sentence. I would suggest the authors to further
 enhance this section, giving a new point of view of these alterations in term of patient
 stratification, as discussed in the later sections

 We agree with the reviewer that this is an important point. A central feature of the binary event
 matrices (BEMs) lies in the fact that the contained cancer functional events (CFEs) are
 rigorously filtered for likely functional alterations in cancer. Therefore, we acquired an
 expanded set of somatic mutations which also contains non-cancer genes, with the updated
 results shown in Fig. S7a:

 **Figure S7: tgDMMRs in the context of genetic alterations, CRISPR screens and drug signatures.**
 (a) A volcano plot summarising associations between tgDMMRs and somatic mutations in cancer cell lines.
 [...]

 Albeit we found a few significant associations, the correlations remain low. Accordingly, we
 revised the corresponding part of the **Results** section:

 “Currently, the majority of biomarkers for patient stratification are genetic alterations,
 thus, we investigated if genetic mutations and copy number alterations may reflect the
 methylation of tgDMMRs. We tested for associations between somatic mutations and

tgdDMRs using linear models (**Methods**). We **only** observed **weak** correlations
between **somatic mutations** and tgdDMRs (FDR < 0.1; **Fig. S7a**).”

Notably, reviewer #1 also suggested expanding the discussion on the absence of key cancer
genes, e.g. *KRAS* lacking associations with dDMRs. Both the absence of many key cancer
genes in the proximity of dDMRs and low correlation with genetic events suggests: (i) only a
few established cancer genes are epigenetically altering drug susceptibilities; and (ii) DNA
methylation likely functions as a complementary mechanism to somatic mutations. For
supporting this notion, we added an example in the **Results** and **Fig. S1f**:

“Among the cancer genes associated with dDMRs, we found that *MGMT* dDMR
methylation in low-grade glioma was associated with response to JQ1 (BET inhibitor;
**Fig. S1f**; $p = 3.7 \times 10^{-5}$, $r = -0.44$). The epigenetic silencing of *MGMT* is frequently
debated as a clinical biomarker³⁰ and previous work revealed that JQ1 disturbs DNA
damage responses by attenuating *MGMT* expression in glioblastoma cells³¹. While
the different treatment responses are often attributed to somatic mutations in cancer
genes, this suggests that DNA methylation can function as a complementary
mechanism.”

**Figure S1:** [...] (f) dDMR in *MGMT* for response to JQ1 in low-grade glioma (LGG).”

Furthermore, we expanded the **Discussion** as the following:

“[...] We observed an enrichment of cancer genes in the proximity of dDMRs, however,
many established cancer genes lacked dDMRs, which suggests that only a minority of
cancer genes may be epigenetically regulated. Furthermore, the modest correlations
with somatic mutations suggest that DNA methylation may function complementary to
genetic alterations for determining cancer drug susceptibilities. [...]”

We also revised the **Methods** section accordingly to reflect the changes:

“Somatic variants and their association with tgdDMRs”

The GDSC project has previously compiled a selection of somatic single nucleotide variants and copy-number alterations ¹¹, which are available at Cell Model Passports (<https://cellmodelpassports.sanger.ac.uk/downloads>). Only somatic mutations in coding regions were considered, which were binarised to represent the mutant and wild type status. Similarly, we binarised amplification and deletions of gene-level copy number alterations. For both we only considered alterations which showed > 3 altered cell lines. For assessing the correlation between genetic alterations and tgdDMRs, we used univariate linear models explaining tgdDMR methylation by the mutational status of each alteration. The p-values were derived from the significance of the regression coefficients and were multiplicity-adjusted by using the Benjamini-Hochberg method.”

REVIEWERS' COMMENTS:

Reviewer #1 (Remarks to the Author):

The authors have addressed questions I raised in the last round. I don't have any further questions.

Reviewer #2 (Remarks to the Author):

All my comments and concerns have been well-addressed!

It also seems to me that the authors have responded and updated the manuscript in a solid manner, according to reviewer #3's comments and questions.